# *GABRG2* Variants Associated with Febrile Seizures

**DOI:** 10.3390/biom13030414

**Published:** 2023-02-22

**Authors:** Ciria C. Hernandez, Yanwen Shen, Ningning Hu, Wangzhen Shen, Vinodh Narayanan, Keri Ramsey, Wen He, Liping Zou, Robert L. Macdonald

**Affiliations:** 1Life Sciences Institute, University of Michigan, Ann Arbor, MI 48109, USA; 2Department of Pediatrics, Seventh Medical Center of Chinese PLA General Hospital, Beijing 100010, China; 3Department of Neurology, Vanderbilt University Medical Center, Nashville, TN 37232, USA; 4Center for Rare Childhood Disorders, Translational Genomics Research Institute, Phoenix, AZ 85004, USA

**Keywords:** GABA_A_Rs, febrile seizures, mutations

## Abstract

Febrile seizures (FS) are the most common form of epilepsy in children between six months and five years of age. FS is a self-limited type of fever-related seizure. However, complicated prolonged FS can lead to complex partial epilepsy. We found that among the GABA_A_ receptor subunit (*GABR*) genes, most variants associated with FS are harbored in the γ2 subunit (*GABRG2*). Here, we characterized the effects of eight variants in the GABA_A_ receptor γ2 subunit on receptor biogenesis and channel function. Two-thirds of the *GABRG2* variants followed the expected autosomal dominant inheritance in FS and occurred as missense and nonsense variants. The remaining one-third appeared as de novo in the affected probands and occurred only as missense variants. The loss of GABA_A_ receptor function and dominant negative effect on GABA_A_ receptor biogenesis likely caused the FS phenotype. In general, variants in the *GABRG2* result in a broad spectrum of phenotypic severity, ranging from asymptomatic, FS, genetic epilepsy with febrile seizures plus (GEFS+), and Dravet syndrome individuals. The data presented here support the link between FS, epilepsy, and *GABRG2* variants, shedding light on the relationship between the variant topological occurrence and disease severity.

## 1. Introduction

GABA, γ-aminobutyric acid, is the primary inhibitory neurotransmitter in the central nervous system (CNS). Although only accounting for 20% of all neurons, GABAergic interneurons are widely distributed in the CNS, sending out broad inhibitory connections throughout the cortex [1,2,3]. GABA_A_ receptors (GABA_A_Rs) are ligand-gated ion channels mainly located at postsynaptic membranes mediating GABA-induced fast inhibitory neurotransmission, controlling neuronal network excitability [1,2,3]. GABA_A_Rs share a conserved structure, as they belong to the Cys loop ligand-activated channel family. These receptors are heteropentameric chloride ion channels composed of two α, two β, and one γ subunit [4,5,6]. Each subunit contains a large N-terminal extracellular domain, followed by four transmembrane segments (M1-M4) and a small C-terminal tail. The subunit interfaces provided by the extracellular domains offer binding sites for GABA, barbiturates, and other allosteric agents; the M2 segments delineate the pore domain; and the large cytoplasmic M3-M4 loop is subject to different types of modulation [4,5,6]. Most postsynaptic GABA_A_Rs are α1β3γ2 receptors, the most abundant GABA_A_R subtype in the CNS [1].

The γ2 subunits play essential roles in receptor trafficking, clustering, synaptic maintenance [2,3], and current kinetic properties [7]. Heterozygous γ2 knockout mice showed a 1/4 reduction in γ2 subunits in the cerebral cortex, hippocampus, and thalamus, accompanied by reduced GABA_A_R clusters and increased anxiety [8]. In addition, defects in the late-stage survival of hippocampal granule cells, including reduced synaptic spine dendritic complexity and maturation, absence-like spike-wave discharges, mild epilepsy, and altered biogenesis of the remaining wild-type γ2 subunits [9,10,11,12,13]. *Gabrg2* deletion in the neocortex and hippocampus caused temperature-dependent seizures in mice and reproduced many features of febrile seizures [14].

*GABRG2* variants have been associated with a broad spectrum of autosomal dominant genetic epilepsy (GEs) syndromes, ranging from the relatively benign FS, childhood absence epilepsy (CAE), and genetic epilepsy with febrile seizures plus (GEFS+), to the more severe epileptic encephalopathies, such as Dravet syndrome, epilepsy with myoclonic-atonic, and Rolandic epilepsy [11,15,16,17,18,19,20,21,22,23,24,25]. The first two GE-*GABRG2* associated variants, *GABRG2(K328M)* and *GABRG2(R82Q)*, were reported in affected individuals of a family with autosomal dominant generalized epilepsy similar to GEFS+, and in affected individuals of a large family with both CAE and FS, respectively. *GABRG2* variants present a wide range of functional deficits that, in most cases, correlate well with the receptor structure–function relationship. The γ2K328M mutation is mapped to the extracellular loop between the M2 and M3 transmembrane domains (M2-M3 loop), a region involved in activating ligand-gated ion channels. Therefore, *GABRG2(K328M)* causes defects in channel kinetics. On the other hand, the conserved γ2R82 residue is critical for forming inter-subunit contacts between the γ subunit positive and β subunit negative sides, consequently preventing incorporation of the γ subunit into the receptor pentamer. Accordingly, *GABRG2(R82Q)* produces dominant negative effects due to ER retention by altered subunit assembly and the reduced surface expression of mutant receptors [18].

Febrile seizures (FS), a type of fever-related seizure, are the most common form of seizures in children between six months and five years of age. Even though most are self-limited, prolonged febrile seizures can lead to complex partial epilepsy. Significant variations in the incidence of FS in different populations and geographic regions strongly suggest a genetic involvement [19]. The cases highlighted above add to the understanding of the basis of FS. However, in all these pedigrees, the γ2R82Q and γ2K328M variants are associated with febrile seizures and extended phenotypes (FS+, GEFS+, and CAE). A proband with a mutation in the γ2 subunit N-terminus (γ2R177G) was also reported to be linked with FS [26]. Recently, two more cases from a cohort of thirty-five patients were described with variants in the γ2 N-terminus associated solely with FS, one of whom inherited the variant from the maternal side (γ2N140S) and the other as a de novo variant (γ2P205H) [27]. Here, we report the crucial γ2 subunit contribution to the FS etiology, as recently acknowledged by Skotte et al. [28], and report a novel cluster of variants harbored in the *GABRG2* gene associated with FS.

In this study, eight *GABRG2* variants (γ2W162X, γ2R215H, γ2R224X, γ2S325L, γ2Y468C, γ2F464S, γ2F457S, and γ2R408K) were found in nine patients (Table 1). GABA_A_R α1 and β3 subunits were co-expressed with wild-type or mutant γ2 subunits in HEK293T cells. Using this heterologous expression system, we found that these variants impaired GABA_A_R biogenesis and channel function to different extents and correlated with mutation-induced perturbations of conserved structural cassettes that mediate receptor function [5,22,24,29,30]. Our findings prove that *GABRG2* variants are a major genetic risk factor for the epileptogenesis of FS and further the understanding of the underlying mechanisms of febrile seizures.

## 2. Materials and Methods

### 2.1. Complementary DNA Constructs

The coding sequences of human α1 (*GABRA1*, NM_000806), β3 (*GABRB3*, NM_021912), and γ2 (*GABRG2*, NM_198904.2) GABA_A_R subunits were subcloned into the pcDNA3.1 expression vectors (Invitrogen, Waltham, MA, USA). Mutant GABA_A_R subunit constructs were generated using the QuikChange site-directed mutagenesis kit (Agilent Technologies, Santa Clara, CA, USA) and confirmed by DNA sequencing.

### 2.2. Cell Culture and Transfection of Human GABA_A_ Receptors

HEK293T cells (ATCC, CRL-11268) were cultured at 37 °C in a humidified 5% CO_2_ incubator and maintained in Dulbecco’s modified Eagle’s medium (Invitrogen) supplemented with 10% fetal bovine serum (Life Technologies, Carlsbad, CA, USA), and 100 IU/mL of penicillin/streptomycin (Life Technologies). For the expression experiments, 4 × 10^5^ cells were transfected using polyethylenimine (PEI) reagent (40 kD, Polysciences, Warrington, PA, USA) at a DNA: to transfection reagent ratio of 1:2.5 and harvested 36 h after transfection. To express WT and variant α1β3γ2 receptors, a total of 3 µg of α1, β3, and γ2 (WT or mutant) subunit cDNAs were transfected at a ratio of 1:1:1 into 60 mm culture dishes. For the mock-transfected condition, an empty pcDNA3.1 vector was added for a final cDNA transfection amount of 3 μg. For the electrophysiology experiments, 2.5 × 10^5^ cells were plated in 100 mm culture dishes and transfected after 24 h with 3 μg of cDNA of each α1, β3, γ2 (wild-type or mutant) subunit, using X-tremeGENE HP DNA transfection Reagent (Roche Diagnostics, Basel, Switzerland), following the manufacturer’s protocol. Recordings were obtained forty-eight hours after transfection.

### 2.3. Determination of GABA-Elicited Responses by Automated Patch Clamp

Whole-cell patch clamp recordings were conducted according to the manufacturer’s standard procedure for the SyncroPatch 384 [31]. Whole-cell voltage-clamp recordings were performed at room temperature on HEK293T cells 48 h after transfection with GABA_A_R subunits, using the SyncroPatch 384PE (Nanion Technologies, Munich, Germany). Cells, at a concentration of 400,000 cells/mL, were harvested in suspension in external solution containing 140 mM of NaCl, 2 mM of CaCl_2_, 4 mM of KCl, 1 mM of MgCl_2_, 5 mM of glucose, and 10 mM of HEPES (pH 7.4 with NaOH, 298 mOsm). The cells were loaded in a multi-hole (4 holes per well) 384-well Nanion patch clamp (NPC) chip with a thick borosilicate glass bottom and an average resistance pore size of ~4 mΩ. Built-in electrophysiological protocols were used, and the data were digitized using PatchControl 384 (Nanion Technologies) application. Using PatchControl 384, electrophysiological parameters, such as seal resistance, capacitance, and series resistance, were determined for each well after the application of a test pulse, and subsequently monitored over time. After the establishment of a whole-cell configuration, the cells were filled with an internal solution containing 10 mM of KCl, 10 mM of NaCl, 110 mM of KF, 10 mM of EGTA, and 10 mM of HEPES (pH 7.2. with KOH, 284 mOsm). The cells were voltage-clamped at −80 mV throughout the duration of the experiment. To minimize channel desensitization and allow repetitive activation, the currents mediated by GABA_A_Rs were activated by a brief exposure (5 ms) to a small GABA volume (2 µL) at the respective experiment concentration, using the ligand puff function in PatchControl 384. Each experiment included GABA activation (repeated three times), two washes in between ligand application for complete ligand washout, the application of a single concentration of ligand to each well, and full activation with 1 mM of GABA (I_MAX_). Therefore, the faster the ligand was applied, the shorter the time required to reach the peak, which was important for precise measurements of maximum currents. For the concentration response (CRC) experiments, a single concentration of GABA was applied to each well (0.1 nM to 1 mM), and maximum activation per well (cell) was determined by a subsequent maximum-effect (EC_MAX_) GABA concentration (1 mM). The currents were normalized to the maximal response of each well, and the concentration–response curves were calculated from single-point additions. The EC_50_ values for each variant or wild-type receptor were calculated across multiple wells for each experimental condition. Each individual GABA concentration was replicated six times per independent experiment, and each experiment was repeated three times (for a total of eighteen replicates per concentration across all independent experiments). However, given the variable proportion of successful patches per condition (i.e., the specific GABA concentration for each WT or variant concentration–response curve), and the nature of the high throughput automated patch clamp technology, an average success rate of 60% was obtained for all experiments. As a result, the number of replicates per experimental conditions varied but on average was approximately eleven. The number of replicate ranges per wild-type or variant analyzed are listed in Section 3.5. To determine the EC_50_ values from the GABA concentration–response curves, a three-parameter sigmoid model that assumes a Hill slope of 1 (Equation (1): Y = Bottom + (Top-Bottom) / (1 + 10^((LogEC50-X))) was used. The bottom plateau was constrained to zero after subtracting the background reference (external solution) signal. The 10–90% rise times and deactivation current time courses were measured from currents obtained by the application of 1 mM of GABA for 5 ms and fitted using the Levenberg–Marquardt least squares method with one-phase exponential function [24,30].

### 2.4. Western Blot and Surface Biotinylation

The HEK293T cells were collected in modified radioimmunoprecipitation assay (RIPA) buffer (50 mM of Tris (pH = 7.4), 150 mM of NaCl, 1% NP-40, 0.2% sodium deoxycholate, 1 mM of EDTA) and 1% protease inhibitor cocktail (Sigma-Aldrich, Saint Louis, MO, USA). The collected samples were subjected to gel electrophoresis using 4–12% BisTris NuPAGE precast gels (Invitrogen) and transferred to polyvinylidene difluoride fluorescence-based (PVDF-FL) membranes (Millipore, Burlington, MA, USA). The primary antibodies used to detect GABA_A_ receptors were as follows: Mouse α1 subunit antibody (1:500; NeuroMab, 75–136), rabbit β3 subunit antibody (1:500; Novus, NB300-199), and rabbit γ2 subunit antibody (1:500; Millipore, AB5559). The Mouse anti-Na+/K+ ATPase antibody (1:000; DSHB, a6F) was used to determine the density of the loading control. IRDye^®^ (LI-COR Biosciences, Lincoln, NE, USA) conjugated secondary antibody was used at a 1:10 000 dilution in all cases. The blotted membranes were scanned using the Odyssey Infrared Imaging System (LI-COR Biosciences). The band-of-interest integrated intensity values were determined using the Odyssey Image Studio software (LI-COR Biosciences). Biotinylation protocols were described previously [23].

### 2.5. Structural Simulation of GABA_A_ Receptor γ2 Subunit Mutations

We used the high-resolution cryo-EM structure of the human pentameric α1β3γ2L GABA_A_ receptor (PDB: 6HUO [6]) along with the Rosetta molecular modeling program [32] to predict GABA_A_R structural rearrangements caused by mutations in the γ2 subunit. The 6HUO coordinates’ entry comprised the full-length human α1 (UniProtKB accession: P14867), β3 (P28472), and γ2L (P18507-2) subunits. The Rosetta-based webtool protocol for stabilizing proteins [33] within the Rosetta software package was accessed within the newest framework for the Rosetta Online Server that Includes Everyone (ROSIE2, https://r2.graylab.jhu.edu/, accessed on 13 January 2023.). The protocol evaluated point mutations and generated models for the mutated structures. The results from the point mutation simulations included models and energy scores for each mutant, which was compared to that of the original sequence and the differences in scores between the native residue and each substitution for a given position. Up to five of the best-scoring structures were generated for each mutation by selecting the parameters recommended by the application. We evaluated mutation-induced structural differences by analyzing the root mean squared (RMS) deviation between the WT structures and the superimposed simulated (mutated) structures. RMS deviation provides carbon-α/carbon-α comparisons between two structurally aligned models; the larger the RMS deviation, the more the mutant structure deviates from the wild-type structure. The structural perturbations in the secondary structure and side chain residues that differ among the WT (in gray) and the mutant simulation (RMS deviation ≥ 0.5 Å) are indicated as a different color from the WT simulation. The figures were rendered using Chimera 1.14 [34] and DeepView 4.1 [35].

### 2.6. Mutation Tolerance of the GABA_A_ Receptor γ2 Subunit

We used the mutation tolerance protocol within the MetaDome 1.0.1 software package (https://stuart.radboudumc.nl/metadome/, accessed on 2 January 2022.) to visualize protein tolerance to genetic variation [36] in the GABA_A_ receptor γ2 subunit. We obtained the variation (non-synonymous over synonymous ratio, dN/dS) at each position in the γ2 subunit (GENCODE: ENST00000356592.3, RefSeq: NM_000816.3, NM_198904.2, UniProtKB: P18507-2) to obtain a tolerance landscape. The reference γ2 subunit transcript was input into MetaDome to retrieve data from gnomAD (https://gnomad.broadinstitute.org/, accessed on 27 January 2022.) [37], pathogenic ClinVar variants (https://www.ncbi.nlm.nih.gov/clinvar/, accessed on 2 January 2022.) [38,39], protein families domain annotations (https://www.ebi.ac.uk/interpro/, accessed on 2 January 2022.) [40], and meta-domain variant annotations. The MetaDome tolerance landscape resulted from computing the dN/dS ratio as a sliding window of 21 residues over the totality of the γ2 subunit (i.e., calculated for ten residues left and right of each residue) [41].

### 2.7. Statistical Analysis

Numerical data were reported as mean ± S.E.M or S.D. Statistical analyses were performed using GraphPad Prism (GraphPad Software 9.4).

## 3. Results

### 3.1. GABRG2 Variants and Proband Phenotypes

Genetic testing was performed in 1411 patients with simple FS, which was defined as generalized seizures lasting less than fifteen minutes and not recurring in twenty-four hours, occurring in probands under five years of age; and epilepsy without acquired factors (e.g., perinatal brain injury, traumatic brain injury, or central nervous system infections). The cases were recruited from the Department of Pediatrics, Seventh Medical Center of the Chinese PLA General Hospital, from 2019 to 2021. Among these 1411 cases, eight probands were identified as carriers of *GABRG2* variants (probands 1 to 8). In addition, proband 9 was recruited from the Center for Rare Childhood Disorders. Variations of *GABRG2* (NM_198904, GRCh37/hg19) were determined by targeted next-generation sequencing of epilepsy (epilepsy gene panel) or whole exome sequencing. Annotation was performed using SAMtools (v0.1.18) and VarScan (v2.3). Variants with a minor allele frequency below 5% (based on 1000 Genomes, dbSNP, EVS, and internal database) were selected. The validation of suspected variants and segregation analysis of both parents was performed using standard post hoc Sanger sequencing.

The clinical characteristics of the nine probands (five females/four males) with *GABRG2* variants were summarized in Table 1. The epilepsy onset was between eleven months, one year, and six months of age in eight probands. The seizure semiology at onset was described as tonic-clonic seizures in four probands, tonic seizures in four probands, complex partial seizures with absence and myoclonic seizures in one proband, and pure febrile seizures in seven probands. The epilepsy outcome was variable, with three probands seizure-free for at least two years (probands 2, 4, and 9) but recurrence by the last follow-up. Physical and neurological examinations were normal in seven probands, and only two probands (patients 1 and 9) experienced developmental delays. Brain MRI was normal in all probands. The EEG showed abnormal activity in four probands. Epileptiform discharges were observed in four probands, including slow-wave and focal slow-wave spikes in three and multifocal slow-wave spikes in one proband. In addition, generalized spike-wave discharges associated with epileptic spasms were observed in one proband. Seven probands were diagnosed with FS with an age of seizure onset of one year. Two probands had complex epilepsy and were diagnosed with GEFS+/FS and MAE.

### 3.2. GABRG2 Variants Were Mapped to Structural Cassettes of the γ2 Subunits That Govern the GABA_A_ Receptor Function

GABA_A_Rs are structurally conserved, with each subunit having well-defined structural domains that contain functional cassettes that determine receptor assembly, trafficking, expression, and activation [4,5,6,42,43]. GABA_A_Rs are hetero-pentameric proteins assembled with γ-β-α-β-α stoichiometry (Figure 1A). GABA_A_R γ2 subunits are composed of a large extracellular N-terminal domain (ECD), followed by four transmembrane (TMD) domains (M1-M4), an extracellular loop (M2-M3 loop), and two cytoplasmic (M1-M2; M3-M4) loops (CD). The *GABRG2* variants identified here were mapped across all three-mayor receptor structural domains (ECD, TMD, and CD) (Figure 1A). In the N-terminal domain, the γ2W162X variant occurred in the β5-β5′ loop in close proximity to a N-glycan linked to α1 subunits at Asn146, which occupy the receptor vestibule (Figure 1B). The γ2R215H and γ2R224X variants neighbored the β8 strand, facing the γ+/β− and α+/γ− interfaces, respectively, and were close to two N-linked glycans observed at the periphery of the β3 (Asn105, β3 strand) and γ2 (Asn247, β9 strand) subunits (Figure 1B). The γ2S325L variant occurred at the M2-M3 loop, which cradles the junctional interface of the N-terminal and transmembrane domains (Figure 1C). The γ2F457S, γ2F464S, and γ2Y468C variants occurred in the transmembrane domain M4 delineating the receptor pore (Figure 1C), while the γ2R408K variant occurred at the M3-M4 cytoplasmic loop (not shown due to lack of structural information). By analyzing the γ1–γ3 subunits’ sequence alignment [38] (Figure 1B,C), we found that positions at W162, S325, F457, F464, and Y468 harbored identical residues across all γ subunits, as well as conserved residues at positions R215 and R224 and at the non-conserved R408 position (not shown). Analysis of the *GABRG2* polymorphism tolerance landscape [36] (Figure 2A) showed that in the γ2 subunit, residues mapped at the ECD that determines agonist binding, and residues mapped at the TMD containing the channel pore had a low tolerance (landscape towards red) to the occurrence of variants and mutations. The γ2 variants and mutations found in these structural domains were considered pathogenic and likely to cause impaired GABA_A_R function. Furthermore, an in silico analysis using Polyphen-2 [44] and SIFT [45] predicted that most of the *GABRG2* variants were likely to be essential for protein function, would not be tolerated, and might disarrange the protein structure (Table 1).

### 3.3. Mutant γ2 Subunits Altered Structural Domains Related to GABA_A_ Receptor Gating

GABA binding to orthosteric sites on the GABA_A_R triggers a cascade of structural changes involving the rearrangement of the N-terminal domain and the movement of transmembrane domains that are transmitted through the pore, and ultimately, results in the alteration of the receptor from an inactive to an active state [4,5,6]. Thus, the structural rearrangements caused by the γ2 variants likely result in conformational changes that could affect ligand binding, channel activation, and receptor biogenesis [46,47,48,49,50,51,52]. To understand the structural changes caused by the γ2 variants, we obtained structural simulations of wild-type and variant GABA_A_Rs using the cryo-EM structure of the human full-length heteromeric α1β3γ2L GABA_A_R (PDB = 6HUO) [6] (Figure 1). We only generated simulations for the γ2 variants product of missense mutations (R215H, S325L, Y468C, F464S, and F457S) that can lead to local effects (intra-subunit) confined to structural domains of the subunit, and global effects (inter-subunit) propagated to a neighboring subunit through rearrangements of nearby structural domains (Figure 2B,C). We calculated the secondary structure rearrangements caused by the presence of the γ2 subunit variant by comparing the root mean square (RMS) deviation between the variant and wild-type structural simulations. When comparisons of the secondary structure and side chain residues showed RMS deviation values ≥ 0.5 Å, these were shown in a color other than gray in the structural simulation as an indication of structural deviation from the wild-type structure (in gray) (Figure 2B,C). Consequently, the γ2R215H subunit variant mapped at the interface between the principal (+) side of the γ2 subunit and the complementary (−) side of the β3 subunit, which delimits the γ2+/β3- interface (Figure 2B,C), produced structural perturbations restricted to the γ2 subunit at the β1 strand, β8 strand, and the loop C (β9-β10 strands). In contrast, the γ2S325L mapped at the M2-M3 loop produced intra- and inter-subunit structural rearrangements, which propagated along the γ2 subunit towards: (1) the ECD via loop 9, loop 2, and the Cys loop; (2) down to the TMD via M1 and M3; and (3) laterally, to the interface between the complementary (−) side of the γ2 subunit and the principal (+) side of the α1 subunit, which delimits the α1+/γ2- interface, to the neighboring α1 subunit via the M2-M3 loop. The γ2Y468C, γ2F464S, and γ2F457S variants lead to TMD-restricted structural perturbations of the γ2 subunit involving M1, M3, and M4, and whose rearrangement progression was determined by the position of the variants at M4. Specifically, γ2Y468C perturbed mainly at the level of the ECD–TMD junction interface, γ2F464S at the middle of the TMD, and γ2F457S towards the intracellular interface of the TMD.

### 3.4. Mutant γ2 Subunits Alter GABA_A_ Receptor Kinetics

Mutations in the receptor activation pathway are expected to affect channel gating [4,5,6,46,47,48,49,50,51,52], as the severity of the loss of function can be attributed to the type of variant (missense versus nonsense). Of the eight *GABRG2* variants studied, six were missense, and two were nonsense. Our simulations determined that γ2 missense variants (γ2R215H, γ2S325L, γ2Y468C, γ2F464S, and γ2F457S) lead to structural changes that span the entire γ2 structure (Figure 2B,C), which perturbed elements involved in the allosteric interactions between ECDs and TMDs initiated through the ligand binding–gating coupling mechanism during receptor activation [4,5,6]. Close examination of the hydrogen bond (H-bonds) network in three γ2 variants mapped to the ECD (γ2R215H), the junctional interface (γ2S325L), and the TMD (γ2Y468C) (Figure 3A), introduced H-bond rearrangements that correlated with the GABA_A_R kinetic deficits observed in the functional studies (Figure 3B,C). As shown, at the ECD, the Arg to His mutation at the residue 215 (γ2R215H) that contributes to the γ2+/β3- interface resulted in the loss of an H-bond with residue D258 in loop C (Figure 3A, top panels). Likewise, the Tyr to Cys mutation at residue 468 (γ2Y468C) resulted in the loss of two H-bonds that stabilize the network between M4 (C468), M1 (Y280), and M3 (A334) in the TMD (Figure 3A, bottom panels). In contrast, the Ser to Leu mutation at the residue 325 (γ2S325L) resulted in the gain of two H-bonds in the Cys loop (L194-F197) and M2 (K324-I321), along with the loss of one H-bond with A322 (M2), and with the neighboring α1K314 side chain rearrangement towards the pore (Figure 3A, middle panels).

Rearrangement of H-bonds in the N-terminal domain and in the transmembrane domain of GABA_A_Rs can cause unfavorable changes in the receptor conformation, leading to a decrease in the response to GABA and decreased activation. We determined current deactivation and activation by measuring the current deactivation time constant and rise time following 5 ms GABA (1 mM) application (Figure 3B,C). Assessment of the GABA_A_Rs kinetic properties showed that most of the γ2 subunit variants significantly (γ2R215H, γ2S325L, γ2F464S, γ2F457S, γ2R408K, γ2R224X) or slightly (γ2Y468C, γ2W162X) slowed receptor activation (Table 2). The deactivation of the receptor was also affected in the same direction. Most of the γ2 subunit variants significantly (γ2Y468C, γ2W162X, γ2R224X), slightly slowed (γ2R215H, γ2S325L, γ2F464S), or did not affect (γ2F457S, γ2R408K) receptor deactivation (Table 2).

The subunit-dependent effects on the kinetic properties appeared to be correlated with the structural domain of the receptor where the variant occurs. Therefore, we examined the kinetic profiles of three of the γ2 variants mapped to the ECD (γ2R215H), the junctional interface (γ2S325L), and the TMD (γ2Y468C) (Figure 3B,C, top panels), which show H-bond rearrangements (Figure 3A). Notably, γ2R215H and γ2S325L preferentially slowed receptor activation, while the γ2Y468C primarily slowed receptor deactivation (Table 2). These results suggested that local rearrangements in the γ2 subunit were transmitted globally to adjacent α1 and β3 subunits, causing changes in the receptor kinetic properties.

### 3.5. Mutant γ2 Subunits Decreased GABA_A_ Receptor Function

GABA_A_R activation and deactivation kinetics can affect GABA potency (EC_50_) and efficacy (I_MAX_), as a result of changes in the rate and duration of channel openings and closures due to variant occurrence. We determined whether FS-associated *GABRG2* variants affect GABA_A_R function by measuring GABA-evoked currents in transfected HEK293T cells following rapid 5 ms 1 mM GABA application (Figure 4A). All γ2 subunit variants significantly reduced the peak amplitude (I_MAX_) of GABA-evoked currents by 50–80% relative to wild-type currents (Figure 4B) (Table 3). The magnitude of the loss of GABA_A_R function caused by the γ2 subunit variant was quantified by calculating the variant maximum current fraction (I_MAX_ variant) compared to the wild type (I_MAX_ WT) (Figure 4C) (Table 3). As a result, the defect in receptor function ranked by current magnitude was γ2R408K > γ2F457F > γ2W162X > γ2R215H > γ2R224X > γ2Y468C > γ2S325L > γ2F464S. The reduction in GABA-evoked currents was negatively correlated with the slower deactivation kinetics elicited by the γ2 variants (Figure 3B). These results suggested that the overall effect caused by the γ2 subunit variants depends on both the potency and efficacy of GABA, as well as the kinetic properties of the activation and deactivation of the mutant receptor.

To gain insight into the impact of *GABRG2* variants on receptor gating, we assessed agonist potency by performing concentration–response relationships (Figure 5). The responses were normalized to the current amplitudes elicited by the maximum GABA concentrations (1 mM) for wild-type (WT) and mutant GABA_A_Rs. Most of the γ2 subunit variants significantly decreased GABA potency (Figure 5A). Mutant γ2 subunits that caused the most significant rightward curve shifts were mapped across the γ2 subunit at the ECD (γ2R215H), TMD (γ2Y468C, γ2F464S, γ2F457S), and CD (γ2R408K). As a measure of how GABA activates mutant GABA_A_Rs, we compared the agonist potency (negative log of EC_50_ or pEC_50_) for WT and mutant receptors (Figure 5B) (Table 4). Accordingly, all missense γ2 subunit variants (γ2R215H, γ2S325L, γ2Y468C, γ2F464S, γ2F457S, γ2R408K) significantly decreased the pEC_50_, while the nonsense γ2 subunit variants (γ2W162X, γ2R224X) had no significant effect. The reduction in GABA potency was positively correlated with the slower activation kinetics elicited by the γ2 variants (Figure 3C). These results demonstrated that the missense γ2 subunits significantly impacted the agonist-induced receptor activation, reducing GABA_A_R currents due to impaired channel gating. In contrast, the “apparent” failure of truncated (nonsense) γ2 subunit variants to affect GABA potency suggests that most GABA_A_Rs on the surface were composed of binary (α1β3) receptors that preserve the agonist binding sites and the receptor activation profile [53].

### 3.6. Mutant γ2 Subunits Changed GABA_A_ Receptor Composition

Structural changes caused by the γ2 variants can also affect receptor stability and its ability to remain on the cell membrane, resulting in decreased or altered surface receptor expression and alterations in subunit composition. To determine whether mutant γ2 subunits changed the composition of GABA_A_Rs trafficked to the surface, we co-transfected HEK293T cells with α1, β3, and wild-type or mutant γ2 subunits at a 1:1:1 α1:β3:γ2 plasmid ratio, and evaluated the surface and total levels of WT α1, β3, γ2, and mutant γ2 subunits by surface biotinylation pull-down experiments. When comparing the expression of GABA_A_Rs containing WT α1, β3, and γ2 subunits with GABA_A_Rs formed by mutant γ2 subunits, the truncated (γ2W162X, γ2R224X) and missense mutants γ2 subunit mapped to M4 (γ2Y468C, γ2F464S, γ2F457S) and CD (γ2R408K) exerted the most significant changes in receptor subunit surface and total levels (Figure 6A) (Table 5). These results demonstrated that all *GABRG2* variants reduce γ2 surface and total levels to different extents. Furthermore, the decrease in γ2 surface levels correlated with the smaller whole-cell currents produced by the mutant γ2 subunits, indicating that these variants reduced GABA_A_R biogenesis.

To further assess how mutant γ2 subunits alter subunit-associated trafficking, we compared the receptor surface to the total expression ratio of wild-type α1, β3, γ2, and mutant γ2 subunits, as a measure of the stoichiometry of GABA_A_Rs found on the cell surface (Figure 6B) (Table 6). Mutant γ2 subunits affected α1 and β3 expression differently, which appeared to be related to the location of the mutation. While “core” structural missense γ2 variants (γ2R215H, γ2F457S, and γ2F464S) mapped to the ECD and TMD domains favored increased α1 and β3 expression, missense γ2 variants mapped to lipid-accessible structural domains (γ2S325L, γ2Y468C, and γ2R408K) resulted in little or no change in the expression of the partnering subunits. In addition, while nonsense γ2R224X and γ2W162X variants increased α1 subunit expression, β3 subunits were increased in γ2R224X, and decreased in γ2W162X.

Our results suggested a correlation between the mutant receptor final stoichiometry expressed at the membrane and its function. Interestingly, the main effect of mutant γ2 subunits appeared to be the reduction in receptor biogenesis. However, *GABRG2* variants also had variable subunit-dependent effects on the activation properties that correlate with the receptor structural domain where the mutation occurs (Figure 2 and Figure 3). To better understand the association between receptor stoichiometry and the change in receptor activation caused by the presence of mutant γ2 subunits, spider diagrams were generated correlating GABA potency (EC_50_) and maximal GABA-evoked current (I_MAX_) (Figure 7A) (Table 3 and Table 4), and the ratio of surface/total expression of α1, β3, and γ2 subunits (Figure 7B) (Table 6). Remarkably, the observed changes in receptor function appeared to correlate with well-defined structural domains recognized as essential to receptor function [46,47,48,49,50,51,52]. Missense mutant γ2 subunits mapped near the GABA orthosteric binding site (γ2R215H), at the interface between the N-terminal domain and the channel pore (γ2S325L), and CD (γ2R408K), reduced GABA potency while maintaining receptor stoichiometry with little or no change in α1 and β3 subunit expression, despite a considerable reduction in mutant γ2 subunits. Furthermore, the incorporation of mutant γ2 subunits resulted in tertiary GABA_A_Rs with decreased GABA-evoked currents and impaired receptor activation (Figure 3 and Figure 4). In contrast, missense mutant γ2 subunits in the pore domain (γ2F457S, γ2F464S, and γ2Y468C) reduced GABA potency by altering receptor stoichiometry with a substantial increase in α1 and β3 subunit expression and a significant reduction in mutant γ2 subunits. Thus, mutant γ2 subunits were incompletely incorporated into ternary α1β3γ2 receptors, with decreased GABA-evoked currents and impaired receptor activation (γ2F457S, γ2F464S) or deactivation (γ2Y468C). In the case of the truncated γ2 variants (γ2W162X, γ2R224X), although they would be expected to exhibit a similar expression profile, they resulted in a mixed behavior. On the other hand, γ2R224X had a greater increase in the surface expression of the α1 subunit relative to β3, and γ2W162X increased both subunits by the same magnitude. However, γ2W162X had a greater increase in total β3 expression compared to surface expression, indicating a negative dominant effect. This would explain the differences in kinetic properties observed, with the slower deactivation caused by γ2W162X, the slower deactivation and activation resulting from γ2R224X, and no differences in GABA potency.

Overall, our results demonstrate that the *GABRG2* variants characterized in this study caused a variety of defects in GABA_A_R function, ranging from structural disruption to conformational changes on the agonist-binding coupling mechanism, disrupting GABA_A_R gating; as well as changes in the receptor stoichiometry that ultimately lead to alterations in the receptor’s response to GABA, and the pharmacology and efficacy of drugs that target the GABA_A_R, potentially affecting the receptor’s physiological role in neuronal excitability.

## 4. Discussion

Variants in *GABRG2* have been associated with a wide range of epileptic syndromes, from mild cases of FS and CAE to more severe forms of epilepsy such as Dravet syndrome [11,15,16,21,22,23,24,25,26,27]. In addition, until recently, a large associative study found a strong association with variants and mutations in the GABA_A_R γ2 subunit and FS [28], confirming that *GABRG2* is clinically relevant and a strong contributor to this pathology.

Thirty *GABRG2* epileptogenic variants were identified in patients with simple FS and various epileptic GE syndromes [15,20,21,22,24,25,54,55,56,57,58,59,60,61,62,63,64,65] (Figure 8) (Table 7). These variants, including those reported in the current study, were selected from the human genome assembly (*GABRG2*, ENST00000639213.2, Ensembl release version 108, accessed on 27 January 2022.) and Humsavar (https://www.uniprot.org/docs/humsavar/, accessed on 27 January 2022.) databases, which include curated variants from ClinVar [39], ExAC, and gnomAD [37]. By the correlation and analysis of genetic information combined with a functional and structural analysis, we established that *GABRG2* variants contribute to the pathogenesis of FS. Most reported *GABRG2* variants were classified as pathogenic (LP/P), and thus, disease-causing (Figure 8A). Furthermore, most variants were mapped to the two major structural domains (ECD and TMD) linked to GABA_A_R activation.

The correlation of *GABRG2* variants by receptor structural topology with epileptic syndrome type determined a phenotype-location profile (Figure 8B), revealing that more severe forms of epilepsy, such as developmental epileptic encephalopathy (DEE), MAE, GEFS+, and FS, had a higher incidence of variants in more than one structural domain than milder forms, such as CAE and generalized tonic-clonic seizures (GTCS). After DEE, FS was the second most common cause of epilepsy, followed by GEFS+. The mapping of all γ2 variants showed that they are distributed throughout the entire subunit (Figure 8C), which ultimately defines, at the structural level, global and local rearrangements that affect receptor gating (Figure 2 and Figure 3). Altering critical structural elements that participate in receptor activation [5,6,48,49,51] correlated well with the GABA_A_R deficits observed in functional studies [15,22,23,24,26].

Furthermore, of the thirty variants included in the analysis, 93% were pathogenic (Figure 8D), and of these, 46% to 54% were found distributed in the extracellular and transmembrane domains (Figure 8E), respectively. As a result, most *GABRG2* variants were located throughout the conduction pathway, leading to defective receptor activation. This is probably the most relevant determinant contributing to the epileptic phenotype. Our functional studies demonstrated significant reductions in GABA_A_R gating due to *GABRG2* variants mapped to the pore domain (γ2F457, γ2F464, and γ2Y468), the coupling zone (γ2S325), and the ECD (γ2R215H) (Figure 3, Figure 4 and Figure 5). Therefore, we demonstrated the pathogenic role of structurally compromised *GABRG2* variants in FS (Figure 2, Figure 3, Figure 4, Figure 5 and Figure 8F).

The interpretation of the effects of missense variants on the receptor function is different from nonsense variants or truncated proteins. The explanation lies in considering that, in general, the subunit that contains the missense variant is always expressed to some degree at the membrane and plays a significant role in receptor function, invariably resulting in defective receptors, and the degree of deficiency will depend on the location of the missense variant. On the other hand, for nonsense variants, the truncated protein is usually retained in the ER; therefore, it is not trafficked or assembled with wild-type subunits and channel function may be seemingly uncompromised, having little to no effect on GABA potency. However, a direct relationship between the expression of different receptor subunits and receptor function is not always straightforward. It is possible to have dissonant apparent GABA potencies with varying and “uncorrelated” receptor cell-surface expression. Thus, nonsense variants result in more α1 and β3 subunit surface expression than their respective total expression, suggesting that the truncated γ2 subunit can exert a gain-of-function effect on the rate of synthesis and degradation of the α1 and β3 subunits. Therefore, more α1 and β3 subunits are expressed on the surface. This may be indicative of the accelerated trafficking of immature (less glycosylated) α1 and β3 subunits on the cell surface, which would also cause a defective membrane-expressed GABA_A_ receptor. This implies that, in this case, the receptor is assembled at the membrane at different subunit stoichiometries compared to the wild type. The α1 and β3 subunits transit to the membrane in greater numbers than the γ2 subunits and are expected to form more binary receptors with variable stoichiometries with a distinct gating. The opposite effect can be also true, where the truncated γ2 subunit can exert a loss-of-function (dominant-negative) effect on the rate of the synthesis and degradation of the α1 and β3 subunits. Consequently, in any case, the overall population of surface receptors will be composed of a repertoire of function-deficient receptors with different stoichiometries.

Our results showed that the *GABRG2* variants identified in patients with FS affected receptor trafficking and gating to different extents. This last statement has been recognized as one of the molecular mechanisms underlying the pathogenicity of *GABR* variants [66,67,68]. However, a clear genotype/phenotype correlation was lacking between *GABRG2* variants and different epileptic phenotypes. Few studies have attempted to establish such correlations [27,63,64,69,70], as a broad phenotypic spectrum is a common feature of genetic epilepsies. Furthermore, key phenotypic differences may even be seen between individuals with the same genetic alteration [71]. In a recent study on *GABRB3* variants, it was reported that there was a significant genotype–phenotype correlation in individuals affected by *GABRB3*-related disorders, as they were clustered in three regions of the β3 subunit that were key to its function (GABA binding region, coupling region, and TMD) [72], clearly linking *GABR* variant pathogenicity and receptor structure–function [29,73,74,75,76,77].

Recent studies suggest the use of benign variants found in the general population as suitable negative controls to establish clear genotype/phenotype correlations [78,79]. Benign variants, as well as disease-associated variants, share the same structural domains important in channel function. Given that GABA_A_Rs belong to the Cys loop ion channel superfamily, which includes glycine, nicotinic acetylcholine (nAChR), and serotonin 5-HT3 receptors, the structure–function correlation of any family member can be inferred from the variant location because substitution in homologous positions results in similar receptor defects [80,81]. In support of this assertion, we previously found that a significant reduction in GABA_A_R function was strongly associated with *GABR* variants mapped within the extracellular and transmembrane domains, which were scored as deleterious in genetic epilepsy patients and individuals from the general population [30,70]. However, the gating defects caused by genetic epilepsy variants, measured as the ratio of activation/deactivation time constants after 10 ms of GABA stimulation, were significantly larger than variants in the general population. Therefore, the presence of *GABR* variants in patients with genetic epilepsies carried a higher risk than those found in the general population. Moreover, a notable difference between the *GABR* variants found in the general population and the *GABR* variants associated with epileptic phenotypes (this study and others [21,24,56,62,67]) was that the former did not produce changes in receptor expression nor in GABA potency [30]. This suggests that *GABR* variants associated with epileptic phenotypes exerted catastrophic effects that had a profound impact on receptor function, proper folding, and stability, potentially leading to impaired receptor signaling. However, many of these studies, including this report, have been limited in validating the variant topology to pathogenicity correlation model by including nonpathogenic variants in the study. It is therefore recommended for future studies to include benign variants in the analysis to increase the robustness of any structure–function inferences.

## 5. Conclusions

Our study adds to the understanding and complements the prevailing paradigm on GABAergic channelopathies caused by the occurrence of *GABRG2* variants. We have expanded the scope of the causal effects of missense and nonsense γ2 subunit variants and the pathogenesis of FS. As established by Skotte et al. [28], the association between *GABRG2* and FS is more explicit. Our study employed a combination of genetic, structural, and functional in vitro assays to unambiguously establish *GABRG2* gene alterations as a genetic risk factor for FS. The next step in discovering new treatments is to establish a holistic model that encompasses the discoveries in bench research and the transfer of this knowledge to the bedside. The structure/function correlation and variant phenotype of additional *GABRG2* variants will undoubtedly nurture further studies on the precise role of γ2 subunits in GABAergic channelopathies and provide new insights into targeted therapy for epileptic syndromes.

## Figures and Tables

**Figure 1 biomolecules-13-00414-f001:**
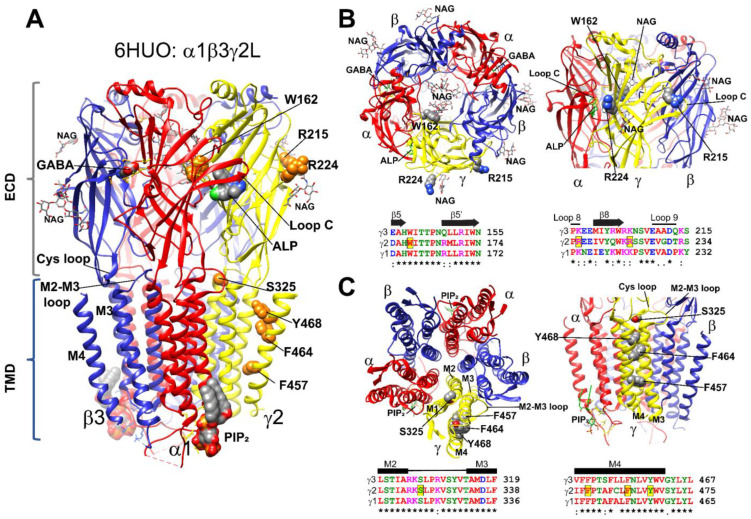
Structural mapping of *GABRG2* variants (**A**) Cryo-EM (PDB#: 6HUO) structure of the GABA_A_ receptor with the β3 subunits in blue, α1 subunits in red, and the γ2 subunit in yellow. γ2 variants were mapped onto the structure and represented as orange space-filled atoms. Binding sites for GABA, alprazolam (ALP), N-linked glycans (NAG), and PIP_2_ are also shown in CPK space-filling atoms. (**B**,**C**) Top-side and bottom-side views of the extracellular (ECD in (**B**)) and transmembrane (TMD in (**C**)) domains of the 6HUO structure represented by ribbons with GABA, ALP, and NAG shown as sticks, and γ2 variants as space-filled representation. Subunit coloring is the same as in (**A**). Sequence alignments of human γ3, γ2 and γ1 GABA_A_R subunits show the position of the variants in the γ2 subunit highlighted in yellow (black box). Residues in the alignments are colored according to their chemical properties. Structural domains related to the secondary structure are represented across subunits above the alignments as β-strands (β5, β5′, β8), loops (8 and 9), and transmembrane domains (M2, M3 and M4). Conserved sites (*) and sites with conservative replacements (:) are indicated below each position of the sequence alignment.

**Figure 2 biomolecules-13-00414-f002:**
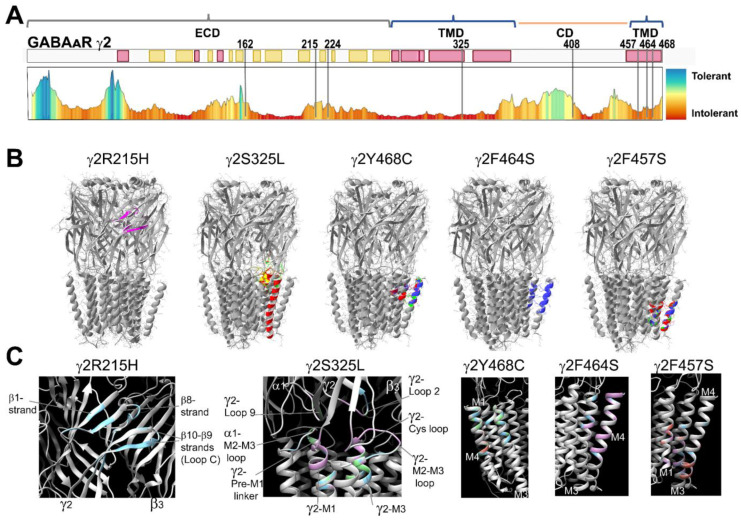
Mutant γ2 subunits induced structural rearrangements in structural domains important for GABA_A_ receptor gating. (**A**) GABA_A_R γ2 tolerance landscape schematic representation of γ2 protein domains that did not tolerate missense variants. As shown, non-tolerated highly pathogenic variations shift the color gradient towards red, whereas tolerated variations with low pathogenicity shift the color gradient towards blue. Elements of the secondary structure of the GABA_A_R γ2 subunit are shown in yellow (β-strands) and red (α-helices) rectangles. Bracket symbols delimit the extracellular (ECD), transmembrane (TDM), and cytoplasmic domains (CD). All GABA_A_R γ2 subunit variants considered in this study were mapped to intolerant landscape regions. (**B**) Global structural perturbations shown on the Cryo-EM 6HUO structure after introduction of single point mutations in the γ2 subunit at positions 215, 325, 468, 464, and 457. The structural rearrangements in the secondary structure and side chain residues that differ among the WT (in gray) and the mutant simulation (RMS deviation ≥ 0.5 Å) are indicated in a different color from the WT simulation. (**C**) Enlarged views of structural domains showing structural rearrangements produced by the γ2R215H, γ2S325L, γ2Y468C, γ2F464S, and γ2F457S mutant subunits. Structural perturbations with RMS deviation ≥0.5 Å are shown and the conformational impact of the γ2 mutations to the α1β3γ2 GABA_A_R are indicated. WT, wild type; RMS, root mean square.

**Figure 3 biomolecules-13-00414-f003:**
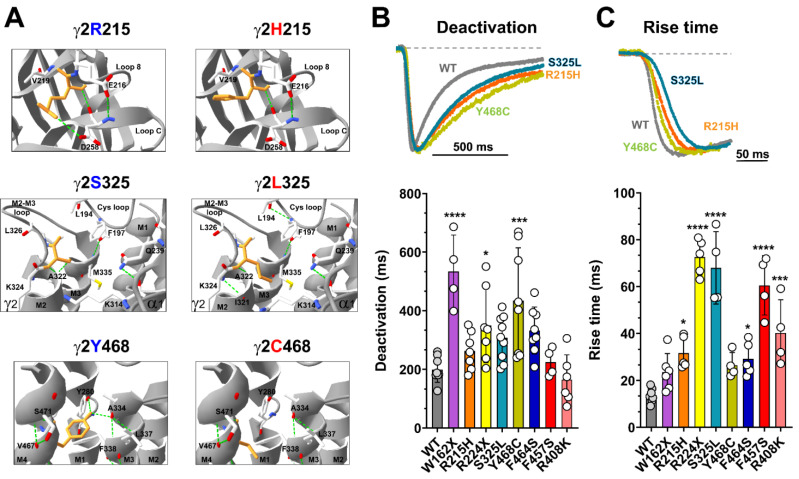
Mutant γ2 subunits altered the kinetic properties of GABA_A_ receptors. (**A**) Close-up views of neighboring residues that were within a 6 Å radius at the site of the mutations corresponding to R215H, S325L, and Y468C in the γ2 subunit structures. The hydrogen bond network of the WT (**left panels**) and mutant structures are shown (**right panels**). Residues corresponding to the location of the substitution are colored orange, and the neighboring residues are colored as follows: carbon, white; oxygen, red; nitrogen, blue. Dashed lines indicate hydrogen bonds. (**B**,**C**) Superimposed representative current traces show deactivation ((**B**), **top panel**) and activation ((**C**), **top panel**) produced by 5 ms of GABA (1 mM) applications to WT (gray) and mutant receptors containing the γ2R215H (orange), γ2S325S (blue), and γ2Y468C (green) subunits. Traces were normalized to WT currents for clarity. Dashed lines represent zero currents. Bar graphs show average deactivation time constant ((**B**), **bottom panel**), and rise time ((**C**), **bottom panel**) from cells co-expressing α1β3 subunits with mutant or WT γ2 subunits. Values are expressed as mean ± S.D (see Table 2). Data points (circles) represent the number of patched wells per experimental condition acquired in three independent experiments. One-way ANOVA with Dunnett’s post-test were used to determine significance. **** *p* < 0.0001, *** *p* < 0.001, and * *p* < 0.05, respectively, relative to WT condition.

**Figure 4 biomolecules-13-00414-f004:**
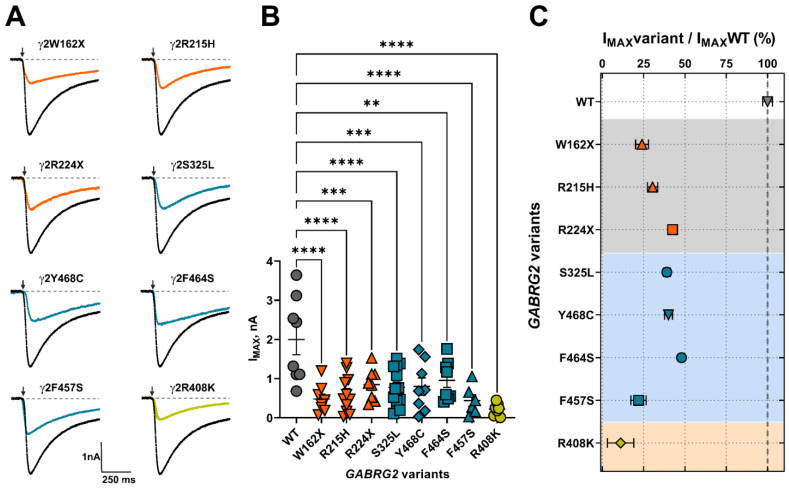
Mutant γ2 subunits decreased GABA-evoked whole-cell currents. (**A**) Representative GABA current traces obtained following rapid application of 1 mM of GABA for 5 ms (arrow). The current traces from GABA_A_Rs containing mutant (in orange red, blue, and green) γ2 subunits were compared to WT (black) current traces. Dashed lines represent zero currents. Activation of GABA_A_ (α1β3γ2) receptors was measured on the SyncroPatch 384PE, and the whole-cell patch methodology and multi-hole NPC chips were used as described in Section 2. (**B**) Bar graphs show the average maximal GABA-evoked peak currents (I_MAX_) for 1 mM of GABA in HEK293T cells co-expressing α1β3 subunits with wild-type (WT) or mutant γ2 subunits. γ2 variant subunit coloring is the same as in (**A**). Numerical data are reported as mean ± S.E.M. Data points represent the number of patched wells per experimental condition acquired in three independent experiments. One-way ANOVA with Dunnett’s post-test was used to determine significance. **** *p* < 0.0001, *** *p* < 0.001, and ** *p* < 0.01, respectively, relative to WT condition. (**C**) I_MAX_ variant over I_MAX_ WT ratios were calculated as estimates of the magnitude of the change in receptor activation by GABA. Panels represent the structural domain location of the γ2 variants (ECD, in gray; TMD, in blue; CD, in orange). Numerical data are reported as mean ± S.D. Some error bars cannot be displayed as they are smaller than the symbol size. A descriptive summary of the data is shown in Table 3.

**Figure 5 biomolecules-13-00414-f005:**
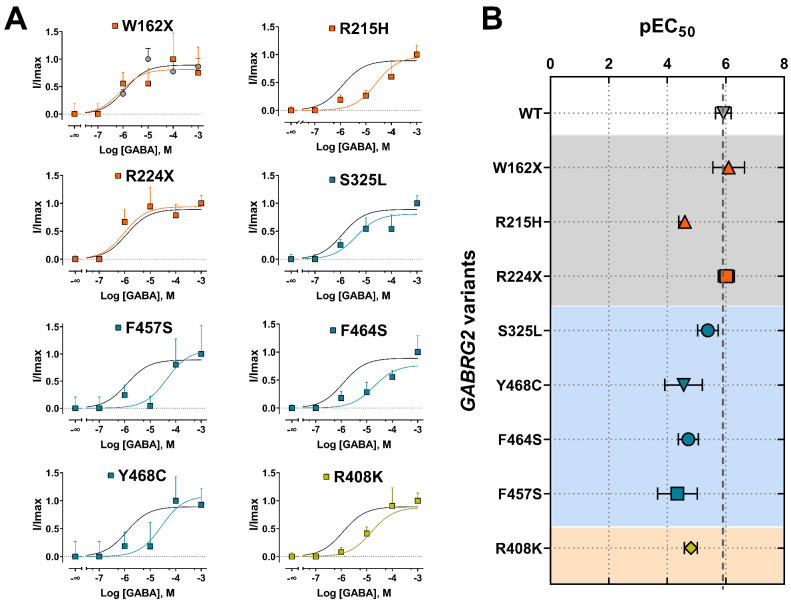
Mutant γ2 subunits altered the potency of GABA_A_ receptors. (**A**) Activation concentration–response curves of GABA_A_ (α1β3γ2) receptors expressed in HEK293T cells on the SyncroPatch 384PE by GABA. A single concentration of GABA was applied to each well. Currents from each well were normalized to the maximal response within the well in the presence of 1 mM of GABA. The data were fitted using a three-parameters sigmoid model, as described in Section 2. The black line represents the fit for the WT condition. The colored lines represent the fit of each experimental condition that corresponds to the data in panel B. Numerical data are reported as mean ± S.E.M. Data points represent the number of successfully patched wells per experimental condition from three independent experiments. (**B**) pEC_50_ (negative Log of the EC_50_) values between WT and mutant γ2 subunits are plotted as the relative change in potency caused by the mutation. Numerical data are reported as mean ± S.D. A summary of the data is shown in Table 4.

**Figure 6 biomolecules-13-00414-f006:**
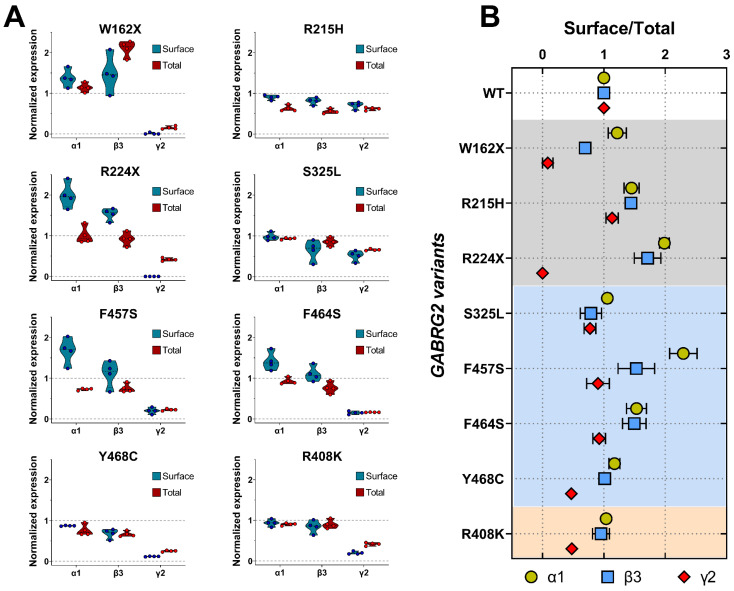
Mutant γ2 subunits changed the composition of GABA_A_ receptors. (**A**) Wild-type or mutant γ2 subunits were co-transfected with α1β3 in HEK293T cells. Surface proteins were biotinylated and pulled down with streptavidin beads. Pulled-down proteins were separated by SDS-PAGE and probed with anti-α1, anti-β3, anti-γ2, or anti-ATPase antibodies. Total cell lysates (pull-down input) were collected, analyzed by SDS-PAGE, and equally blotted by anti-α1, anti-β3, anti-γ2, and anti-ATPase antibodies. Surface and total expression of GABA_A_Rs containing mutant γ2 subunits were normalized to WT α1β3γ2 subunits. Numerical data are reported as mean ± S.E.M. A summary of the data is shown in Table 5. (**B**) Surface/total expression ratios were determined as the fraction of surface to total altered GABA_A_R subunit, as an indicator of the magnitude of the negative or positive effect of the presence of mutant γ2 subunits. Numerical data are reported as mean ± S.E.M. Some error bars cannot be displayed as they are smaller than the symbol size. The data were summarized in Table 6.

**Figure 7 biomolecules-13-00414-f007:**
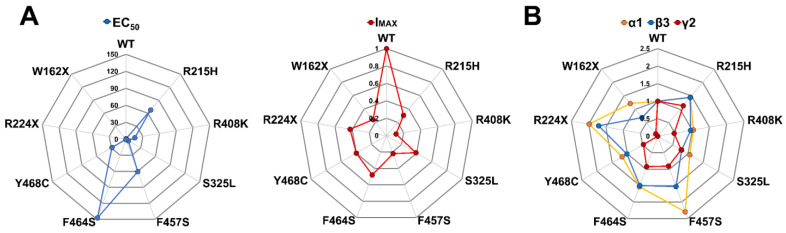
GABA_A_ receptor impairment profile caused by mutant γ2 subunits. Spider plots of the GABA_A_R expression and function compare the impairment magnitude caused by mutant γ2 subunits. (**A**) The EC_50_ and I_MAX_ of GABA_A_R containing mutant γ2 subunits were normalized against WT. The data are related to the assays shown in Figure 4 and Figure 5. (**B**) Surface and total expression of GABA_A_R co-expressing WT α1, β3, γ2, and mutant γ2 subunits were normalized against WT. These data are related to the assays shown in Figure 6. Values other than 1 indicate a loss (<1) or gain (>1) effect.

**Figure 8 biomolecules-13-00414-f008:**
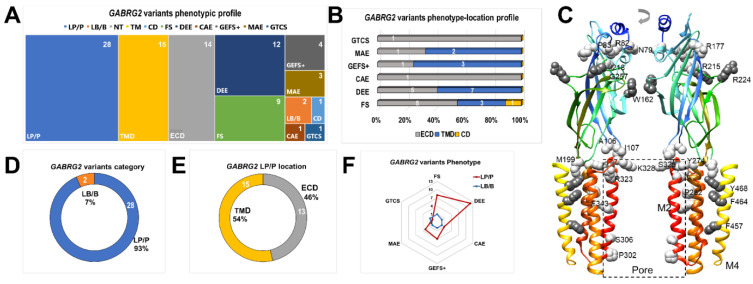
*GABRG2* variants contributed to the pathogenesis of FS. (**A**) Comparison of the phenotypic profile of thirty *GABRG2* variants reported in this and previous studies (Table 7). *GABRG2* variants are grouped by structural domain location (ECD, extracellular domain; TMD, transmembrane domain; and CD, cytoplasmic domain), epilepsy phenotype (FS, febrile seizure; DEE, developmental epileptic encephalopathy; CAE, childhood absence epilepsy; GEFS+, genetic epilepsy with febrile seizures plus; GTCS, generalized tonic-clonic seizure; and MAE, myoclonic-astatic epilepsy), and pathogenicity (LP/P, likely pathogenic/pathogenic; and LB/B, likely benign/benign). The *GABRG2* variants’ distribution in each category was assigned a number ranging from 1 to 28. Note that a specific variant may be represented in more than one category. (**B**) Occurrence of thirty *GABRG2* variants by structural location (ECD, TMD, and CD) and epileptic phenotype. The number inside the bars corresponds to the variants included in that category. Variants that cause the most complex epileptic phenotypes occur in more than one GABA_A_R structural domain. (**C**) Side view of the extracellular and transmembrane domains of two opposing γ2 subunits represented as rainbow ribbons. γ2 subunit variants are displayed in dark (the present study) and light gray (previously reported). In addition, key structural domains are indicated. (**D**,**E**) Doughnut plots classify *GABRG2* variants by pathogenicity (LP/P, n = 28, and LB/B, n = 2) and structural location (NT, n = 13, and TM, n = 15, LP/P only), respectively. The plots show most variants are pathogenic, localizing preferentially to two key receptor activation domains. (**F**) Spider diagram comparing the pathogenicity and type of epilepsy profile of *GABRG2* variants. The features of the thirty *GABRG2* variants are summarized in Table 7.

**Table 1 biomolecules-13-00414-t001:** Genetic and clinical features of patients carrying *GABRG2* variants.

	Patient 1	Patient 2	Patient 3	Patient 4	Patient 5	Patient 6	Patient 7	Patient 8	Patient 9
Inheritance	Father	De Novo	Father	Mother	De Novo	Mother	De Novo	Mother	Father
Genomic position *	Chr5:161580337	Chr5:161580316	Chr5:161530907	Chr5:161580349	Chr5:161580349	Chr5:161580169	Chr5:161576165	Chr5:161524801	Chr5:161530933
Gene	*GABRG2*	*GABRG2*	*GABRG2*	*GABRG2*	*GABRG2*	*GABRG2*	*GABRG2*	*GABRG2*	*GABRG2*
Nucleotide change	c.1391T > C	c.1370T > C	c.644G > A	c.1403A > G	c.1403A > G	c.1223G > A	c.974C > T	c.485G > A	c.670C > T
Amino acid change	p.Phe464 Ser	p.Phe457 Ser	p.Arg215 His	p.Tyr468 Cys	p.Tyr468 Cys	p.Arg408Lys	p.Ser325 Leu	p.Trp162 X	p.Arg224 X
Predictions Polyphen2	Damaging	Damaging	Damaging	Damaging	Damaging	Benign	Damaging	Damaging	Damaging
Predictions SIFT	Deleterious	Deleterious	Deleterious	Deleterious	Deleterious	Benign	Deleterious	Deleterious	Deleterious
Sex	Male	Female	Male	Male	Male	Female	Female	Female	Female
Diagnosis	FS	FS	FS	GEFS+/FS	FS	FS	FS	FS	MAE
Current age	Two years one month	Five years nine months	One year one month	Four years seven months	Three years three months	Four years three months	One year five months	Two years seven months	Seven years
Age at onset	One year three months	One year three months	One year	Eleven months	One year six months	One year six months	One year five months	One year	N/D
EEG	The bilateral posterior has a slightly higher quantity of slow waves in the waking stage; bilateral central area episodic spikes and wave complex during sleep.	Background activity lower frequency. Right central vertex spikes and slow-wave complex discharges; widespread slow-wave paroxysm during sleep.	Normal	A few diffuse spike-wave and slow spike-wave discharges during sleep. Posterior head (three years); normal EEG (three years nine months).	Normal	Normal	N/D	Normal	Epileptiform discharges: synchronous bilateral spike–slow wave and bifrontal maximum consistent (3.8 yrs.); generalized 2.5 Hz spike-wave complex discharge associated with epileptic spasms and slow rhythms (four years); generalized spike-wave discharges in stage II sleep (seven years).
Seizure types	Tonic-clonic seizures	Tonic-clonic seizures	Tonic seizures	Tonic seizures	Tonic seizures	Tonic-clonic seizures	Tonic-clonic seizures	Tonic seizures	Complex partial seizures; frontal lobe epilepsy; absence and myoclonic seizures.
AED response	N/A	N/A	N/A	LEV was only used as a prophylactic drug in febrile conditions.	CZP was only used as a prophylactic drug in febrile conditions.	N/A	N/A	N/A	Responded to ketogenic diet therapy; and TPM.
Seizure outcome	Six times per year	Twice and seizure-free for three years	Two times total, within two months	Seizure-free for two years and nine months	Two times per year	Three times per year	Two times total, within one month	Five times per year	At 4 years old, 3 times per mon.; at 6.5 years old, seizure-free for a while, then returned.
MRI findings	CT: normal	N/D	Normal	Normal	Normal	Normal	CT: normal	Normal	Within normal limits.
Neurological exam	Normal	Normal	Normal	Normal	Normal	Normal	Normal	Normal	Shaking extremities; epileptic spasms showed spells consisting of a head nod, arms abduction, and recovery within 3–4 s associated with spike-wave discharges on EEG (4 years).
Motor development	Normal	Normal	Normal	Normal	Normal	Normal	Normal	Normal	N/D
Cognitive outcome	Intellectual disability	Normal	Normal	Normal	Normal	Normal	Normal	Normal	Mild intellectual developmental delay.
Others	Sleep disorders	Neonatal hypoxic-ischemic encephalopathy	Has family history, brother.	Has family history, six cases out of ten family members, including three males and three females.	N/A	N/A	Neonatal amniotic fluid choking cough.	Has family history, mother.	Headaches (3–4 times a month) in the last 2.5 years.

*GABRG2* (NM_198904). ***** Genomic positions are related to the GRCh37/hg19 human genome assembly. FS, febrile seizures; GEFS+, genetic epilepsy with febrile seizures plus; MAE, myoclonic-astatic epilepsy; AEDs, antiepileptic drugs; TPM, topiramate; LEV, levetiracetam; CZP, clonazepam; N/A, not applicable; N/D, non-disclosure.

**Table 2 biomolecules-13-00414-t002:** Effects of *GABRG2* variants on α1β3γ2 receptor macroscopic kinetics.

	Deactivation (ms)					Rise Time (ms)				
	Mean	S.D.	n	Summary	Adjusted *p* Value ^1^	Mean	S.D.	n	Summary	Adjusted *p* Value ^1^
WT	199.9	43.88	8			13.67	3.065	7		
W162X	533.8	124.6	4	****	<0.0001	23.51	7.964	6	ns	0.3086
R215H	263	64.65	7	ns	0.7903	31.68	6.894	5	*	0.0123
R224X	350.1	126.4	7	*	0.0414	72.64	7.629	5	****	<0.0001
S325L	307.7	75.67	9	ns	0.1888	68.01	15.44	4	****	<0.0001
Y468C	433.9	180.7	8	***	0.0002	26.52	5.408	4	ns	0.1744
F464S	332.6	79.92	8	ns	0.0748	29.22	7.173	5	*	0.0398
F457S	224.8	48.26	4	ns	0.9994	60.51	12.64	4	****	<0.0001
R408K	165.4	84.56	6	ns	0.9924	40.23	14.13	4	***	0.0003

^1^ One-way ANOVA with Dunnett’s multiple comparisons test was used to determine statistical significance among *GABRG2* variants with respect to the wild-type (WT). Number of replicates ranged from 4 to 9 successfully patched wells per experimental condition acquired across three independent experiments. ns, no significance. **** *p* < 0.0001, *** *p* < 0.001, and * *p* < 0.05, respectively, relative to WT condition.

**Table 3 biomolecules-13-00414-t003:** Effects of *GABRG2* variants on α1β3γ2 receptor macroscopic GABA-evoked currents.

	I_MAX_ (nA)					I_MAX variant_/I_MAX WT_ (%)				
	Mean	S.E.M.	n	Summary	Adjusted *p* Value ^1^	Mean	S.D.	n	Summary	Adjusted *p* Value ^1^
WT	2.00	0.38	8			100.00	2.95	8		
W162X	0.48	0.13	8	****	<0.0001	24.07	3.87	8	****	<0.0001
R215H	0.61	0.14	11	****	<0.0001	30.44	3.10	11	****	<0.0001
R224X	0.85	0.14	8	***	0.0007	42.52	1.70	8	****	<0.0001
S325L	0.78	0.14	13	****	<0.0001	39.07	2.13	13	****	<0.0001
Y468C	0.80	0.22	8	***	0.0004	40.06	2.39	8	****	<0.0001
F464S	0.96	0.18	8	**	0.0025	47.93	1.58	8	****	<0.0001
F457S	0.44	0.15	6	****	<0.0001	21.96	4.60	6	****	<0.0001
R408K	0.22	0.06	7	****	<0.0001	11.10	7.98	7	****	<0.0001

^1^ One-way ANOVA with Dunnett’s multiple comparisons test was used to determine statistical significance among *GABRG2* variants with respect to the wild type (WT). Number of replicates ranged from 6 to 13 successfully patched wells per experimental condition acquired across three independent experiments. **** *p* < 0.0001, *** *p* < 0.001, and ** *p* < 0.01, respectively, relative to WT condition.

**Table 4 biomolecules-13-00414-t004:** Effects of *GABRG2* variants on α1β3γ2 receptor GABA potency.

	pEC_50_	S.D.	n	Summary	Adjusted *p* Value ^1^
WT	5.92	0.27	5–10		
W162X	6.10	0.54	5–14	ns	0.2353
R215H	4.60	0.21	5–16	****	<0.0001
R224X	6.02	0.27	5–11	ns	0.8233
S325L	5.39	0.35	5–13	****	<0.0001
Y468C	4.56	0.64	5–12	****	<0.0001
F464S	4.72	0.34	5–10	****	<0.0001
F457S	4.35	0.68	6–13	****	<0.0001
R408K	4.81	0.22	5–10	****	<0.0001

^1^ One-way ANOVA with Dunnett’s multiple comparisons test was used to determine statistical significance among *GABRG2* variants with respect to the wild type (WT). Nonlinear regression with log(agonist) vs. response (three parameters) equation was used to determine pEC_50_. The range of successful patches (replicates) across three independent experiments is indicated under the “n” column. ns, no significance. **** *p* < 0.0001 relative to WT condition.

**Table 5 biomolecules-13-00414-t005:** Effects of *GABRG2* variants on α1β3γ2 receptor expression.

	Surface														
	α1					β3					γ2				
	Mean	S.E.M.	n	Summary	Adjusted *p* Value ^1^	Mean	S.E.M.	n	Summary	Adjusted *p* Value ^1^	Mean	S.E.M.	n	Summary	Adjusted *p* Value ^1^
WT	0.998	0.003	4			0.998	0.003	4			0.997	0.002	4		
W162X	1.373	0.110	4	*	0.0249	1.481	0.232	4	**	0.0018	0.177	0.049	4	****	<0.0001
R215H	0.901	0.030	4	ns	0.9688	0.809	0.044	4	ns	0.5488	0.701	0.044	4	ns	0.1179
R224X	1.992	0.158	4	****	<0.0001	1.528	0.076	4	***	0.0005	0.000	0.000	4	****	<0.0001
S325L	0.987	0.047	4	ns	0.9999	0.652	0.127	4	*	0.0464	0.513	0.066	4	**	0.0018
Y468C	0.871	0.004	4	ns	0.8831	0.680	0.060	4	ns	0.0805	0.118	0.003	4	****	<0.0001
F464S	1.416	0.114	4	**	0.0094	1.107	0.094	4	ns	0.9413	0.152	0.018	4	****	<0.0001
F457S	1.669	0.163	4	****	<0.0001	1.110	0.162	4	ns	0.9331	0.200	0.038	4	****	<0.0001
R408K	0.936	0.043	4	ns	0.9972	0.841	0.079	4	ns	0.7358	0.198	0.017	4	****	<0.0001
	**Total**														
	**α1**					**β3**					**γ2**				
	**Mean**	**S.E.M.**	**n**	**Summary**	**Adjusted *p* Value ^1^**	**Mean**	**S.E.M.**	**n**	**Summary**	**Adjusted *p* Value ^1^**	**Mean**	**S.E.M.**	**n**	**Summary**	**Adjusted *p* Value ^1^**
WT	0.997	0.002	4			0.995	0.003	4			0.998	0.003	4		
W162X	1.143	0.056	4	ns	0.1496	2.099	0.100	4	****	<0.0001	0.152	0.019	4	****	<0.0001
R215H	0.630	0.038	4	****	<0.0001	0.561	0.029	4	****	<0.0001	0.622	0.018	4	****	<0.0001
R224X	1.012	0.108	4	ns	0.9997	0.923	0.083	4	ns	0.8255	0.417	0.018	4	****	<0.0001
S325L	0.934	0.011	4	ns	0.8971	0.855	0.052	4	ns	0.1798	0.661	0.011	4	****	<0.0001
Y468C	0.759	0.068	4	**	0.0031	0.669	0.033	4	****	<0.0001	0.251	0.008	4	****	<0.0001
F464S	0.932	0.040	4	ns	0.8822	0.763	0.071	4	**	0.0042	0.164	0.002	4	****	<0.0001
F457S	0.728	0.013	4	***	0.0006	0.757	0.054	4	**	0.0031	0.226	0.012	4	****	<0.0001
R408K	0.902	0.015	4	ns	0.5654	0.904	0.053	4	ns	0.6213	0.416	0.019	4	****	<0.0001

^1^ Two-way ANOVA with Dunnett’s multiple comparisons test was used to determine statistical significance among *GABRG2* variants with respect to the wild-type (WT). ns, no significance. **** *p* < 0.0001, *** *p* < 0.001, ** *p* < 0.01, and * *p* < 0.05, respectively, relative to WT condition.

**Table 6 biomolecules-13-00414-t006:** Effects of *GABRG2* variants on α1β3γ2 receptor surface trafficking.

	Surface/Total														
	α1					β3					γ2				
	Mean	S.E.M.	n	Summary	Adjusted *p* Value ^1^	Mean	S.E.M.	n	Summary	Adjusted *p* Value ^1^	Mean	S.E.M.	n	Summary	Adjusted *p* Value ^1^
WT	0.9997	0.0002	4			0.9995	0.0003	4			0.9997	0.0002	4		
W162X	1.2186	0.1474	4	ns	0.7537	0.6958	0.0822	4	ns	0.4171	0.0861	0.0861	4	****	<0.0001
R215H	1.4511	0.1230	4	ns	0.0832	1.4422	0.0371	4	ns	0.0931	1.1348	0.0998	4	ns	0.9731
R224X	1.9858	0.0804	4	****	<0.0001	1.7130	0.2173	4	**	0.0012	0.0000	0.0000	4	****	<0.0001
S325L	1.0567	0.0504	4	ns	0.9996	0.7890	0.1730	4	ns	0.7855	0.7743	0.0963	4	ns	0.7279
Y468C	1.1723	0.0907	4	ns	0.9051	1.0146	0.0715	4	ns	0.9999	0.4699	0.0037	4	*	0.0277
F464S	1.5324	0.1621	4	*	0.0265	1.4955	0.1905	4	*	0.0453	0.9243	0.1031	4	ns	0.9994
F457S	2.2951	0.2244	4	****	<0.0001	1.5292	0.2986	4	*	0.0277	0.9053	0.1843	4	ns	0.9968
R408K	1.0362	0.0309	4	ns	0.9997	0.9543	0.1379	4	ns	0.9996	0.4772	0.0377	4	*	0.0308

^1^ Two-way ANOVA with Dunnett’s multiple comparisons test was used to determine statistical significance among *GABRG2* variants with respect to the wild type (WT). ns, no significance. **** *p* < 0.0001, ** *p* < 0.01, and * *p* < 0.05, respectively, relative to WT condition.

**Table 7 biomolecules-13-00414-t007:** Attributes of selected *GABRG2* variants found in epileptic cases.

*GABRG2* Variant	Variant Category ^1^	Phenotype ^2^	Location ^3^	Reference	gnomAD ^4^	Other SNPs ^4^
R82Q	LP/P	CAE	ECD	[54,56,57]	yes	R82W ^4^
R82Q	LP/P	FS	ECD	[54,56,57]	yes	-
K328M	LP/P	GEFS+	TMD	[58]	no	-
R177G	LP/P	FS	ECD	[59]	no	R177Q ^5^
N79S	LB/B	GTCS	ECD	[15,60,61]	yes	-
P83S	LP/P	FS	ECD	[15,62]	no	P83L ^6^
P83S	LP/P	DEE	ECD	[63]	no	-
Y274C	LP/P	MAE	TMD	[55]	no	-
R323Q	LP/P	DEE	TMD	[21,24]	yes	-
R323Q	LP/P	MAE	TMD	[20]	yes	-
A106T	LP/P	DEE	ECD	[24,64]	yes	A106P ^5^
I107T	LP/P	DEE	ECD	[24]	no	-
P282S	LP/P	DEE	TMD	[24]	no	-
R323W	LP/P	DEE	TMD	[24]	yes	-
F343L	LP/P	DEE	TMD	[24]	no	-
P282T	LP/P	DEE	TMD	[63]	no	-
S306F	LP/P	DEE	TMD	[63]	no	-
G257R	LP/P	DEE	ECD	[21]	no	-
I218S	LP/P	DEE	ECD	[65]	no	-
M199V	LP/P	GEFS+	ECD	[25]	yes	M199K/T ^7^
P302L	LP/P	DEE	TMD	[22]	no	-
F464S	LP/P	FS	TMD	Current study	no	-
F457S	LP/P	FS	TMD	Current study	no	F457L ^5^
R215H	LP/P	FS	ECD	Current study	no	R215P/L ^4^
Y468C	LP/P	GEFS+	TMD	Current study	yes	-
Y468C	LP/P	GEFS+	TMD	Current study	yes	--
R408K	LB/B	FS	CD	Current study	no	R408T ^5^
S325L	LP/P	FS	TMD	Current study	no	-
W162X	LP/P	FS	ECD	Current study	no	-
R224X	LP/P	MAE	ECD	Current study	no	R224Q ^5,7^

^1^ LP/P, likely pathogenic/pathogenic; and LB/B, likely benign/benign. ^2^ FS, febrile seizure; DEE, developmental epileptic encephalopathy; CAE, childhood absence epilepsy; GEFS+, genetic epilepsy with febrile seizures plus; GTCS, generalized tonic-clonic seizure; and MAE, myoclonic-astatic epilepsy. ^3^ ECD, extracellular domain; TMD, transmembrane domain; and CD, cytoplasmic domain. ^4^ Ensembl release 108-Oct 2022 (Transcript: ENST00000639213.2 *GABRG2*-215). ^5^ gnomAD. ^6^ ClinVar. ^7^ ExAC.

## Data Availability

All data generated or analyzed during this study are included in this published article.

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
