# Peer review of "GABRG2* Variants Associated with Febrile Seizures"

_biomolecules, 2023, doi:10.3390/biom13030414_

Round 1
Reviewer 1 Report
This article reports several epilepsy-associated variants in the GABRG2 gene encoding for the gamma2 subunit of the GABAA variants. The authors have examined the predicted pathogenicity and evaluated the expression levels and functional changes caused by the receptor. They have also aggregated some of the data on 37 variants including those that are published here, and those in the literature.
Overall, the paper has several major issues, both technical and with respect to the broader literature, that need to be resolved prior to publication.
Major issues:
1. The functional values including EC50, change in EC50, Imax and Normalized Surface expression levels need to be summarized in a table so that the reader can easily compare the results that are being described.
2. There is no statistical analysis of the EC50 values or the surface expression levels. Without this there is insufficient rigour for publication in the scientific literature.
3. Correct use of statistical tests. The authors use a parametric ANOVA with Dunnett’s post-hoc test to determine changes in the Imax. The author’s need to justify their choice of a parametric test which would be suitable for comparing logEC50 values, but not EC50 values, for instance. If the authors do wish to use the Dunnett’s post-hoc test, then the sample size for the wild-type values too are low for ANOVA with Dunnett’s post-hoc test. The sample size should not be the same as the variants but derived from the equation n(wt) = n(variants) x sqrt (number of variants). (e.g. If you test 8 n’s of 9 variants, you need an n = 24 wild-types (https://www.jstor.org/stable/2528490.)
4. Fig 4B purports to show EC50 variant/EC50 wt. I strongly doubt this to be the case, for instance R215H appears to have shifted nearly 2 log units. It looks to be logEC50 variant/ logEC50 wt, but that’s not a useful mathematical transformation. Instead the authors should define the change in EC50 as a log shift with the equation log10(EC50 variant/EC50 wt) or logEC50 variant – logEC50 wt, which is the same thing.
5. It is unclear whether Figure 4 is displaying sem or sd? In any case, plotting the standard deviation would be more appropriate.
6. More detail is required on how EC50’s in Fig 4 were determined. Were the EC50’s determined for individual experiments, and then a mean +/- sem derived? If so, the logEC50 +/- sem should be derived as the normal distribution is around the log scale. If not, please explain how.
7. The concentration-response curves do not appear very well done at all. The decision to only use one GABA concentration per log unit should be argued why, as many other groups would measure two or 3 (0.1, 0.3,1… or 0.1, 0.2, 0.5, 1…). Looking at the curves, I would judge the EC50 values from these experiments to be poor approximates and not truly measure the effect of the variant on the concentration-response curves. This is important as several authors have recently published or in pre-prints claim a critical importance of the EC50 on the phenotype of the variant in other GABA receptors, including the concept that increases or decrease in this value are associated with different phenotypes. (Hannan et al https://doi.org/10.1101/2021.12.08.471533, Chen et al PMID: 35269865, Gallagher et al https://journals.sagepub.com/doi/full/10.1177/15357597221130199, Absalom et al PMID: 35269865). These need to be substantially improved to be publication quality.
8. There is a lack of negative controls, particularly in this specific case where amino acid variants at the same amino acid in the sequence but with a different amino acid change, are present in gnomAD. It is absolutely essential to assess a negative control and determine that it is the same as wild-type to demonstrate the absence of a systematic error. I advise the following gnomAD variants need to be performed: F457L and R408T that are in the gnomad database and the variant.
9. Fig6A. This is misleading, there are three comparisons here and they are all lumped together in the diagram. Please change the diagram to discretely show the different comparisons.
10. Fig 6B should either be changed so that the bar graphs display raw numbers, or the total number of patients in brackets is placed on the axis.
11. The spider plots add very little value, what’s the point of them when the Imax and EC50 values are related, as too are the subunit expression values?
12. The results are internally inconsistent. Take F464S. The alpha and beta subunits are expressed at the cell surface at the same level as the wild-type, but the gamma subunit is barely detectable. The concentration-response curve is shifted enormously to the right. Yet, the maximum currents are half the wild-type and are, in fact higher than any other variants including the S325L variant that has no change in the EC50, and much higher surface expression levels. Can that be explained? In cases where there is reduced expression of the gamma containing receptors on the cell surface, why is the EC50 not shifted to the left as it is in the truncation variants? The author’s need to provide an explanation to this, as it doesn’t make sense.
13. The genotype/phenotype correlation needs more information. Several residues are shown that have variants in the gnomAD database but it is not clear whether these are the same amino acid variants, or different ones. Please clarify and comment on the pathogenicity of variants that appear in the gnomAD database and that you are using for the genotype phenotype correlation.
14. The conclusion refers to discovering new treatments. In this situation where GABAA receptors can be targeted, I would have thought a comprehensive evaluation of the efficacy of present treatments is also a high priority.
15. The self-citations are quite high at 36% of all references and, as mentioned before, not a complete description of the field at the current time.
Minor
Spell and grammar check required.
Variant and mutant are used interchangeably throughout the document, best change all to variant.
Author Response
We thank the editor and reviewers for their comments and suggestions. We appreciate the reviewers' critiques which have raised the scientific quality of our study. Accordingly, we have improved the manuscript based on these comments. We provide a point-by-point response to the reviewers’ comments below.
Reviewer #1
- The functional values including EC50, change in EC50, Imaxand Normalized Surface expression levels need to be summarized in a table so that the reader can easily compare the results that are being described. There is no statistical analysis of the EC50 values or the surface expression levels. Without this there is insufficient rigour for publication in the scientific literature. Correct use of statistical tests. The authors use a parametric ANOVA with Dunnett’s post-hoc test to determine changes in the Imax. The author’s need to justify their choice of a parametric test which would be suitable for comparing logEC50 values, but not EC50 values, for instance. If the authors do wish to use the Dunnett’s post-hoc test, then the sample size for the wild-type values too are low for ANOVA with Dunnett’s post-hoc test. The sample size should not be the same as the variants but derived from the equation n(wt) = n(variants) x sqrt (number of variants). (e.g. If you test 8 n’s of 9 variants, you need an n = 24 wild-types (https://www.jstor.org/stable/2528490.) Fig 4B purports to show EC50 variant/EC50 I strongly doubt this to be the case, for instance R215H appears to have shifted nearly 2 log units. It looks to be logEC50 variant/ logEC50 wt, but that’s not a useful mathematical transformation. Instead the authors should define the change in EC50 as a log shift with the equation log10(EC50 variant/EC50 wt) or logEC50 variant – logEC50 wt, which is the same thing. It is unclear whether Figure 4 is displaying sem or sd? In any case, plotting the standard deviation would be more appropriate. More detail is required on how EC50’s in Fig 4 were determined. Were the EC50’s determined for individual experiments, and then a mean +/- sem derived? If so, the logEC50 +/- sem should be derived as the normal distribution is around the log scale. If not, please explain how. The concentration-response curves do not appear very well done at all. The decision to only use one GABA concentration per log unit should be argued why, as many other groups would measure two or 3 (0.1, 0.3,1… or 0.1, 0.2, 0.5, 1…). Looking at the curves, I would judge the EC50 values from these experiments to be poor approximates and not truly measure the effect of the variant on the concentration-response curves. This is important as several authors have recently published or in pre-prints claim a critical importance of the EC50 on the phenotype of the variant in other GABA receptors, including the concept that increases or decrease in this value are associated with different phenotypes. (Hannan et al https://doi.org/10.1101/2021.12.08.471533, Chen et al PMID: 35269865, Gallagher et al https://journals.sagepub.com/doi/full/10.1177/15357597221130199, Absalom et al PMID: 35269865). These need to be substantially improved to be publication quality.
Response: We really appreciate the reviewer's complete and thorough revision, and in light of the reviewer’s comments, we have a better understanding of the manuscript’s shortcomings. In particular by not elaborating on the data statistical analysis in our first submission. To address this very valid concern and to clarify all the issues related to the statistical treatment of the data, the manuscript has been supplemented with 4 additional tables (Tables 2 to 5):
Table 2. Effects of GABRG2 variants on α1β3γ2 receptor macroscopic GABA-evoked currents. Related to figures 2 and 5.
Table 3. Effects of GABRG2 variants on α1β3γ2 receptor GABA potency. Related to figures 3 and 5.
Table 4. Effects of GABRG2 variants on α1β3γ2 receptor expression. Related to figures 4 and 5.
Table 5. Effects of GABRG2 variants on α1β3γ2 receptor surface trafficking. Related to figures 4 and 5.
These tables summarize the number of data points analyzed per experimental condition, the statistical treatment applied by case, and the interpretation given to each of the treatments in question. Each table describes the data that is represented in Figures 2 through 5. Regarding the statistical test chosen to analyze the data, we would like to acknowledge the reviewer’s concern about analyzing non-Gaussian distributed data with a parametric test. We are very grateful to the reviewer for pointing this out, and as a result, we have changed our data presentation regarding EC50 values to negative log or pEC50 values, As a result, what was originally a treatment of non-parametric geometric mean has been transformed to the (rightfully) accepted arithmetic mean and Gaussian distribution resulting from this data treatment. With pEC50 values at hand, it was possible to analyze the data with the standard one-way (or two-way) ANOVA and Dunnett's test. The text was modified accordingly to reflect these changes. On the other hand, we established that the optimum and most cost-effective number of points used for the construction of the GABA-CRCs was 6 in order to record all the variants on the same plate and experimental day. The elevated cost of SynchroPatch plates makes it prohibitive to do otherwise and we have seen that six-concentration curves yield identical results to even twelve-concentration curves. Given the spread of the concentrations used, it was not possible to use half-log or even smaller dilutions and still cover the full spectrum of the GABA response. However, only five points were plotted because the vehicle control was omitted (as is often done) because it was used as a data normalizing parameter. This approach ensured less errors due to differences in transfection, culture health, growth rate, etc.
- There is a lack of negative controls, particularly in this specific case where amino acid variants at the same amino acid in the sequence but with a different amino acid change, are present in gnomAD. It is absolutely essential to assess a negative control and determine that it is the same as wild-type to demonstrate the absence of a systematic error. I advise the following gnomAD variants need to be performed: F457L and R408T that are in the gnomad database and the variant. The genotype/phenotype correlation needs more information. Several residues are shown that have variants in the gnomAD database but it is not clear whether these are the same amino acid variants, or different ones. Please clarify and comment on the pathogenicity of variants that appear in the gnomAD database and that you are using for the genotype phenotype correlation.
Response: The reviewer raises an interesting point. However, most studies published to date focus on common genetic variations, generally defined as having a minor allele frequency (MAF) of at least 5%, while rare variants are defined as 0.1% ≤ MAF ≤ 1%. In Table 6 (Table 6. Attributes of selected GABRG2 variants found in epileptic cases), we list the thirty GABRG2 variants selected for the comparison of the occurrence of variants in key receptor structural domains and epileptic phenotypes. We analyzed the GABRG2 variants using the Ensembl database released in Oct-2022 that contains the most recent annotated data of the human genome for the gamma-aminobutyric acid type A receptor subunit gamma2 (Source:HGNC Symbol;Acc:HGNC:4087). We used ENST00000639213.2 as a reference. This transcript has 10 exons, is annotated with 36 domains and features, and is associated with 23897 variant alleles. In Table 6, we annotated the natural variants found in it and the source used with their respective reference in case the variant is associated with a specific epileptic phenotype. By analyzing the different variants present in the Ensembl database, only six GABRG2 variants causative of an epileptic phenotype (R82Q, N79S, R323Q, R323W, M199V, and Y468C) have been described in normal population cohorts. The MAF for these variants is <0.01, which classifies them as ultra-rare (MAF < 0.1%) variants present in the normal population. The occurrence of a different amino acid substitution at the same position has been reported in several of the GABRG2 variants as indicated in Table 6, which suggest that the variants occurred in locations of the protein sequence described as “hot spots” for variation where receptor structural domains are important for function (see tolerance landscape in Figure 1A). Natural variants at these positions, when not correlated to any epileptic phenotype, argue in favor of the ultra-rare nature (MAF<0.01) and the presence of additional factors (age, race, population size, geography, gene dosage, penetrance, susceptibility, etc.) contributing to the final morbidity outcome. The natural variants F457L and R408T suggested by the reviewer as “negative controls” have no known disease-associated variants, fall into the same MAF category as ultra-rare variants (MAF < 0.1%), and have a low “disease propensity” value of 0.85, according to the VarSite database (https://www.ebi.ac.uk/thornton-srv/databases/VarSite), which annotates known disease-associated variants in human genes with structural information from the Protein Data Bank (PDB). Thus, from a structural standpoint, both natural variants appear to have less harmful effect on the receptor function. We strongly believe that using these variants as "negative controls" does not have any advantage because the occurrence of multiple natural variants in non-essential regions has no effect. In addition, the native protein is already a variant of sorts with a very high occurrence rate. On the other hand, even if these “less deleterious” variants have some effect, this does not negate our findings compared to controls. We respectfully disagree with the reviewer in that these would constitute an appropriate negative control as they may, “have an effect”. As mentioned elsewhere, the complex association between subunit stoichiometry, surface expression, and response to GABA does not correlate in a straightforward manner.
- The genotype/phenotype correlation needs more information. Several residues are shown that have variants in the gnomAD database but it is not clear whether these are the same amino acid variants, or different ones. Please clarify and comment on the pathogenicity of variants that appear in the gnomAD database and that you are using for the genotype phenotype correlation.
Response: Complementing what we answer above, our study identified eight FS-causing GABRG2 variants from a small cohort of patients with epilepsy without acquired factors, and these variants were detected by the diagnostic epilepsy panel or whole exome sequencing. The small sample size limits the correlation between clinical history and genetic testing, a one-to-one correlation. To unravel the genotype-phenotype correlations and the possible variants that modify the clinical presentation in FS, one could eventually perform genome-wide association studies (GWAS), which are an important tool to identify common variants associated with complex diseases. However, these require the inclusion of more than five hundred thousand to two million single nucleotide polymorphisms (SNPs) (Ha et al., 2014) in order to evenly cover the human genome, which is beyond the scope of the present study. Although this is an inherent limitation of our study (and other published ones), we still consider it important and valuable to show the predicting power of these variants in establishing the function of the different receptor structural domains. In fact, this approach has been validated by us and others (ACS Chem. Neurosci. 2021, 12, 3, 562–572; Front. Mol. Neurosci., 20 November 2020, https://doi.org/10.3389/fnmol.2020.602559; ACS Chem. Neurosci. 2021, 12, 13, 2421–2436; ACS Chem. Neurosci. 2022, 13, 21, 3044–3056; Nature Communications volume 11, Article number: 5369 (2020)).
- This is misleading, there are three comparisons here and they are all lumped together in the diagram. Please change the diagram to discretely show the different comparisons.Fig 6B should either be changed so that the bar graphs display raw numbers, or the total number of patients in brackets is placed on the axis.
Response: The figure has been modified to clarify the different correlations made. In addition, Table 6 (Attributes of selected GABRG2 variants found in epileptic cases) has been added to the manuscript describing the phenotypic features of the 30 GABRG2 variants selected.
- The results are internally inconsistent. Take F464S. The alpha and beta subunits are expressed at the cell surface at the same level as the wild-type, but the gamma subunit is barely detectable. The concentration-response curve is shifted enormously to the right. Yet, the maximum currents are half the wild-type and are, in fact higher than any other variants including the S325L variant that has no change in the EC50, and much higher surface expression levels. Can that be explained? In cases where there is reduced expression of the gamma containing receptors on the cell surface, why is the EC50not shifted to the left as it is in the truncation variants? The author’s need to provide an explanation to this, as it doesn’t make sense.
Response: We acknowledge the reviewers' comments and agree that a direct relationship between the expression of different receptor subunits and receptor function is not always straightforward. Taking F464S and S325L as examples, we observe significantly higher expression of the α1 and β3 subunits (140% and 110%), compared to γ2, with only 15%. for F464S. This implies that in this case, the receptor is assembled at the membrane at different subunit stoichiometries compared to the wild type. The α1 and β3 subunits transit to the membrane in greater numbers than the γ2 subunits and are expected to form more binary receptors with variable stoichiometries with a distinct gating. In regard to this, for each γ2 subunit trafficked to the membrane, the assembled receptor will consist of all three subunits, i.e., a tertiary receptor, showing poor activation because the mutant subunit in TM4 affects the pore domain. Consequently, the overall population of surface receptors will be composed of a repertoire of activation-deficient binary or tertiary receptors with different stoichiometries. This cell-surface expressed GABAR heterogeneity plausibly accounts for the dextrally-shifted poor activation seen for the GABA CRC. In contrast, for S325L, the expression of the α1 subunit (98%) is similar to that of wt, but β3 (63%) and γ2 (50%) are reduced. This case suggests that when the stoichiometry of membrane-trafficking receptors is not as compromised as seen for F464S, tertiary receptors would constitute the largest population at the surface. However, since β3 is also reduced, suggesting a dominant negative mutant effect, there is less tertiary receptor trafficking to the membrane. In addition, since the mutation is in TM2, specifically in the receptor coupling zone, gating is affected, accounting for the GABA-CRC dextral shift. As exemplified above, the interpretation of the effects of missense mutations on receptor activation is quite different from nonsense mutations or truncated proteins. The explanation lies in considering that, in general, the subunit that contains the missense mutation is always expressed to some degree at the membrane and plays a significant role in receptor activation, invariably resulting in defective receptors, and the degree of deficiency will depend on the location of the missense mutation. On the other hand, for nonsense mutations, the truncated protein is usually retained in the ER and therefore not trafficked or expressed with wild type subunits and channel activation may be seemingly uncompromised little to no effect on GABA potency. We hope the reviewer will recognize how it is possible to have dissonant apparent GABA potencies with varying and “uncorrelated” receptor cell-surface expression.
- The conclusion refers to discovering new treatments. In this situation where GABAA receptors can be targeted, I would have thought a comprehensive evaluation of the efficacy of present treatments is also a high priority.
Response: We thank the referee for the comments and agree that it is a great point of discussion. Massive studies of thousands of individuals in the normal population are very useful in trying to infer the association of genes commonly found or reported in epileptic syndromes such as SCN1A, SCN8B, GRIN2B, SCN2A, CACNA1A, GRIN1, SCN1B, and KCNMA1, but they lose precision for genes that have little association with specific epileptic syndromes. The sub classification of epilepsies in the different types and subclasses of syndromes (FS, CAE, GTCS, Infantile Spams, Lennox Gastaut Syndrome, Dravet syndrome, Otahara syndrome, MEA, etc), reduces the inference and correlation with new genes that, until recently, had not been reported in the literature as causing the disease (e.g., GABAA receptors). Our studies propose that, in a small sample (in this case, 1411 individuals), it is possible to infer structure and function from the correlation between patient phenotypes and variant “severity”. Our study enables the implementation of personalized medicine given the “predictive” nature of diagnosing the presence of the variant in a given patient. Nevertheless, to address a comprehensive evaluation of the efficacy of AED treatments against GABR genes associated epileptic syndromes, we need a comparison of large-scale studies as reported by Skotte and collaborators (Brain. 2022 Apr 18;145(2):555-568). But because the backgrounds of the studied population are so diverse, sometimes it is difficult to establish a direct correlation (similar observation made above in regard of natural variants) between “genotype” and “phenotype” as suggested by the reviewer. The number of cases reported in the present study was limited to a select number of patients, where the diagnosis was individualized and rigorously established following clinical protocols for FS, and where no other gene associated with FS was found.
- The self-citations are quite high at 36% of all references and, as mentioned before, not a complete description of the field at the current time.
Response: Additional references were added accordingly in order to provide a more balanced state of the field. Nonetheless, we respectfully hope that the reviewer acknowledges our significant contributions to this field, justifying the citations made in the manuscript. In this case, we think that rather than being self-serving, the bibliography cited reflects the current landscape for the field which includes our contributions.
Minor
- Spell and grammar check required.
Response: We apologize for any spelling or grammar mistakes incurred. We reevaluated the text and hope this has been fully mitigated.
- Variant and mutant are used interchangeably throughout the document, best change all to variant.
Response: We thank the reviewer for this valid observation. The appropriate changes have been applied.
Reviewer 2 Report
The paper aims at providing evidence that genetic mutations of gamma2 subunit of native GABA receptors are involved in epileptic seizures. To this end, the contribution of mutated gamma2 subunits to receptor activity is analyzed in HEK cell lines.
Trafficking of GABA receptor subunits during status epilepticus is well documented (see for instance Goodkin et al, 2008, J Neuroscience 28, 2527-2538). In the present study the experimental design aims at providing new additional evidence on the topic, in order to develop a better treatment of the disease. The experimental design is accurate, results are convincing.
However, I have problems with the presentation of data. Fig 1-3 are clear; in Fig 2B and 3B some mutations, lack error bars, as shown in Fig 2A and 3A. Fig 4a is less informative (at least the cartoon representation of results is confusing). In detail, several mutations produce surface expression of alpha and beta subunits greater than their respective total: could that be better explained? As for Fig 4b, it is unreadable without at least a color legend
If possible, Table 1 should be rearranged for easier reading. In the text age of onset is reported within the first year of life (lines 150-151), but data in the Table are 1 year or more?
Minor corrections
1) References should be corrected according to journal standards. List of authors in Ref 13 is incomplete. In Materials and Methods the SyncroPatch reference (line 108) is 41: clearly a mistake.
2) English usage and spelling should be checked. Same are as follows: line 51: bold? line 126: nlotted?; line 239: missense? line 273: to with? Lines 302-303: bold?
Author Response
Reviewer #2
- However, I have problems with the presentation of data. Fig 1-3 are clear; in Fig 2B and 3B some mutations, lack error bars, as shown in Fig 2A and 3A. Fig 4a is less informative (at least the cartoon representation of results is confusing). In detail, several mutations produce surface expression of alpha and beta subunits greater than their respective total: could that be better explained? As for Fig 4b, it is unreadable without at least a color legend
Response: We appreciated the reviewers’ comments. Accordingly, the figures have been modified. In the case of no error bars appearing for certain points, this is due to the unfortunate occurrence where the symbol size is larger than the error range, resulting in the errors being “hidden” behind the symbols. A comment has been added to the figure legend when this was the case in hopes of clarifying to the readership, that error bars have been taken into consideration and are shown whenever possible. We still stand by this approach as we think that clearly showing symbols in a graph supersedes the need to visualize small error spreads. A table summary accompanying the figures was also added to further clarify the data descriptive statistics.
As for the observation that some variants seem to result in more α1 and β3 subunits surface expression than their respective total expression, suggests that the variant in the γ2 subunit exerts a gain-of-function effect on the rate of synthesis and degradation of the α1 and β3 subunits. Therefore, more α1 and β3 subunits are expressed on the surface. This may be indicative of accelerated trafficking of immature (less glycosylated) α1 and β3 subunits to the surface, which would also cause a defective membrane expressed GABAA receptor.
- If possible, Table 1 should be rearranged for easier reading. In the text age of onset is reported within the first year of life (lines 150-151), but data in the Table are 1 year or more?
Response: We thank the reviewer for noting this discrepancy. We have modified the text in the table to better reflect the in-line text.
Minor corrections
- References should be corrected according to journal standards. List of authors in Ref 13 is incomplete. In Materials and Methods, the SyncroPatch reference (line 108) is 41: clearly a mistake.
Response: We thank the reviewer for highlighting these mistakes. The text was corrected to show the corresponding reference number.
- English usage and spelling should be checked. Same are as follows: line 51: bold? line 126: nlotted?; line 239: missense? line 273: to with? Lines 302-303: bold?
Response: We apologize for the persistence of errors and spelling mistakes. We revised the manuscript and correct errors accordingly.
Round 2
Reviewer 1 Report
It is with regret that I recommend rejection of the manuscript.
The concentration-response curves are simply of inadequate quality for publication. Specifically, the F457S, F464S and Y468C cannot be used to generate EC50 values. The use of one concentration per log unit is simply inadequate to generate the quality of data required for the study and the cost of plates as a reason not to perform the experiments adequately is not something as a reviewer that I am willing to accept. A minor point, in the table the authors should make it clear whether the n's are individual concentration response curves or individual data points.
The second major flaw is the lack of a negative or benign control. The importance of performing these controls are very well explained by Brnich et al (PMID: 31892348). This control ought to be standard across functional assays by now and yields important information on the quality of assay being performed and the ability to successfully predict pathogenicity from an assay. I do not find the response of the author an adequate reason not to include benign variants in the assays, regardless of whether they are my suggestion or those chosen by the authors.
Author Response
Reviewer 1:
- It is with regret that I recommend the rejection of the manuscript
We are disappointed that the reviewer felt our latest revisions fell short of the reviewer's expectations. We apologize for any ambiguity that led to the misinterpretation of the main ideas and scientific content we wanted to convey in our manuscript. Regarding this, and as a continuation of prior work, we aimed to functionally characterize a subset of naturally-occurring missense and nonsense variants mapped to the γ2 GABAAR subunit and their correlation with clinical severity. Instead of offering a “classification” for the described variants (since this, in many instances, has been done by others), we instead rely on the pathogenic severity to infer structural functionality. In line with this, we, and others, have successfully employed this approach to (1) - assign functional and mechanistic properties to different structural domains and (2) - predict the association between variant topology and disease severity. It is worth pointing out the power of this approach which, in all instances, has allowed us to model and assign mechanistic modules to structural domains to be later validated by cryo-EM structural work. We highlight this assertion to stress the validity of this approach,
Further, we acknowledge the reviewer’s concerns and appreciate the insightful comments regarding the data statistical treatment and the use of benign variants as negative controls. Based on the reviewer’s comments, we made substantial modifications to the manuscript that certainly improved clarity. For this revision, we performed additional revisions of the data analysis and supplemented this with a more detailed description of the methods. For example, (1) - The baseline (vehicle) control was included in the concentration-response curves to improve the accuracy of the non-linear regression fits. (2) - The experimental design was further clarified, and correspondingly, the figure legends now include the technical replicate “n” range for the curves depicted in Figures 2, 3, and Table 3, as well as the number of independent experiments performed. Thus, we cordially ask the reviewer to kindly consider these novel changes in hopes that this latest version addresses any prior concerns.
- The concentration-response curves are simply of inadequate quality for publication. Specifically, the F457S, F464S, and Y468C cannot be used to generate EC50 values. The use of one concentration per log unit is simply inadequate to generate the quality of data required for the study, and the cost of plates as a reason not to perform the experiments adequately is not something as a reviewer that I am willing to accept. A minor point, in the table, the authors should make it clear whether the n's are individual concentration-response curves or individual data points.
We acknowledge the reviewer's concerns regarding the suitability of the concentration or dose-response curves. We recognize the deficiencies of the initial versions, and we are incredibly grateful for the insightful suggestions and comments. Nonetheless, we respectfully disagree with the reviewer’s assertion about the data adequacy in support of the conclusion reached. We believe that the following criteria should be met when performing dose-response curves: (1) – the presence of well-defined bottom and top plateaus, (2) – sufficient data points to define the sigmoid midpoint inflection, (3) – sufficient technical replicates to provide enough “power” to accurately determine the midpoint inflection (i.e., EC50 values). We also respectfully disagree with the reviewer and their belief that the inclusion of inter-log GABA concentrations are necessary criterion if criteria 1 through 3 are met. We cannot look away from the fact that more data points would increase the robustness of the curves, and to make up for this perceived deficiency, we employed the standard three-parameter sigmoid model that assumes a Hill slope equal to 1 and fixed the bottom plateau to zero after subtracting the background or reference signal in the presence of the external solution (vehicle). We want to respectfully remind the reviewer that our analysis aimed to rank the deviation of the variants from the wild-type subunit GABA response and not to provide a comprehensive analysis of the actual agonist response profile for each variant. We understand that this might be seen as a missed opportunity, but we understand that providing more detailed data just goes beyond the scope of our manuscript (please refer to the points raised in the response above). In a previous response, we noted that our “ranking” aim guided our experimental design and determined our choice to include all the variants in a single Nanion chip plate (NCP). With 384 wells available for the variants analyzed in this study and the inclusion of vehicle and maximum-response references and the necessary replicates to account for quality-control-well exclusion (i.e., due to low seal resistance or capacitance, no catch disabled wells, unstable seal or seal loss among other factors imposing a 40% exclusion rate), the possibility of including different agonist concentrations was limited.
Moreover, in our hands, adding extra data points to GABA dose-response curves regularly results in comparable results to those obtained with more coarse curves. Our experimental design represents a sensible, cost-effective compromise that must be considered in light of the scope of this study. We apologize for the lack of clarity in previous responses and regret not making this point clear. As a result of the revised analysis, we modified the corresponding tables and figures (please refer to the reaction above) to account for any (minor) changes caused by the latest analysis and expanded the methods section to clarify procedures that were unclear in previous sections.
- The second major flaw is the lack of a negative or benign control. The importance of performing these controls is very well explained by Brnich et al. (PMID: 31892348). This control ought to be standard across functional assays by now and yields important information on the quality of the assay being performed and the ability to successfully predict pathogenicity from an assay. I do not find the response of the author an adequate reason not to include benign variants in the assays, regardless of whether they are my suggestion or those chosen by the authors.
We appreciated the review's point of view regarding the use of benign variants as “negative control.” However, we respectfully disagree that such an approach would modify our conclusions. We maintain our belief that our study design appropriately addresses the scope of the study. At this point, we think it is essential to acknowledge that the reviewer has highlighted an important, albeit not crucial (the root of our disagreement) limitation of our study. Nonetheless, we think it is essential to acknowledge this limitation for the potential readership of our study so that this weakness is well understood. To address this, we added two additional paragraphs at the end of the discussion section to explain this limitation.
Reviewer 2 Report
Thanks to authors for their responses. .Figures are now clearer, and adding the actual values (Tables) nicely completes the presentation of data
References still need revision. In detail: Refs 1- to 36, plus Ref 39 should be corrected. Ref 14 is incomplete. Refs 37-38 and 40 to 54 are ok
In “Conclusions” the reference to Skotte is wrong: 24, but should be 25
I am sorry to be so demanding, but I think that small inaccuracies would spoil an otherwise well organized paper.
I recommend the paper for publication
Author Response
Reviewer 2:
- Thanks to the authors for their responses. Figures are now clearer, and adding the actual values (Tables) nicely completes the presentation of data.
We appreciate the reviewer's valuable and insightful comments. We are also grateful for the time and effort invested into revising our manuscript.
- References still need revision. In detail: Refs 1- to 36, plus Ref 39 should be corrected. Ref 14 is incomplete. Refs 37-38 and 40 to 54 are ok
We thank the reviewer for pointing out these deficiencies. The references have been corrected.
- In “Conclusions,” the reference to Skotte is wrong: 24, but it should be 25
This mistake has been corrected.
- I am sorry to be so demanding, but I think that small inaccuracies would spoil an otherwise well-organized paper.
We appreciate the high bar set forth by the reviewer. The reviewer’s comments have significantly improved the manuscript.
Round 3
Reviewer 1 Report
I don't really see that there is a misinterpretation of the main ideas or scientific content. There are two main features to the manuscript, the description and functional characterization of the novel variants and the association between variant topology and disease severity. I completely agree these are valid approaches and the premise of the manuscript, the importance to science and human health is not in question here. The question is whether the conclusions reached by the authors are justified by the data presented, and whether the experimental design conforms to the current standards in the field.
The additional paragraphs in the discussion on the use of negative controls improves the document substantially, and the reference to Hernandez et al 2016 is important as pathogenic and normal population control variants are compared. Cell surface expression of population control differences do not show a significant difference to the wild-type, and should the authors not incorporate negative controls into their assays, this information should be provided in the manuscript to give the reader more confidence in this result. Alternatively, the maximum currents show 9/22 variants with reduced currents, and although significant with a Fisher's Exact Test, the false positive rate here is quite high. This information should be provided in the manuscript at the point of the GABA currents, so that the reader understands the limitations of the experiments. The concentration-response curves have not been compared with benign variants as far as I can tell, and that should be incorporated into the manuscript.
As for the concentration-response curves, I am still confused with the statistical analysis. A range is given for the number of experiments, but you can't do the ANOVA and post-hoc test with a range of values, it needs to be a specific number of samples. How was this done and is this the correct test to be doing with the experimental setup? If EC50 values and errors were not derived from individual cells or groups of cells, but instead from curvefitting, then surely this is not the correct test?
Author Response
Reviewer 1
Comments and Suggestions for Authors
1-I don't really see that there is a misinterpretation of the main ideas or scientific content. There are two main features to the manuscript, the description and functional characterization of the novel variants and the association between variant topology and disease severity. I completely agree these are valid approaches and the premise of the manuscript, the importance to science and human health is not in question here. The question is whether the conclusions reached by the authors are justified by the data presented, and whether the experimental design conforms to the current standards in the field.
Response 1: We appreciate the reviewers’ comments and acknowledgement of the importance of our study.
2-The additional paragraphs in the discussion on the use of negative controls improves the document substantially, and the reference to Hernandez et al 2016 is important as pathogenic and normal population control variants are compared. Cell surface expression of population control differences do not show a significant difference to the wild-type, and should the authors not incorporate negative controls into their assays, this information should be provided in the manuscript to give the reader more confidence in this result. Alternatively, the maximum currents show 9/22 variants with reduced currents, and although significant with a Fisher's Exact Test, the false positive rate here is quite high. This information should be provided in the manuscript at the point of the GABA currents, so that the reader understands the limitations of the experiments. The concentration-response curves have not been compared with benign variants as far as I can tell, and that should be incorporated into the manuscript.
Response 2: An additional paragraph has been added to the discussion section of the manuscript to address this point and highlight the limitations of our study. This point has been expanded in the latest version to address the reviewer’s concern.
3-As for the concentration-response curves, I am still confused with the statistical analysis. A range is given for the number of experiments, but you can't do the ANOVA and post-hoc test with a range of values, it needs to be a specific number of samples. How was this done and is this the correct test to be doing with the experimental setup? If EC50 values and errors were not derived from individual cells or groups of cells, but instead from curvefitting, then surely this is not the correct test?
Response 3: We would like to clarify this point and acknowledge the lack of clarity in the initial version of the manuscript. To perform the ANOVA and post hoc test, we used the resulting pEC50 values from each independent experiment. Hence the number of independent experiments for each condition was 3. In the current “Methods” section and in the figure legends we explain that the dose-response curves are the result of a range of data points for each concentration. This is an imposition of the high throughput method utilized the obtain the curves. The percent success rate for obtaining stable patches along the SynchroPatch chip is inherently variable and impossible to control. Curtailing the number of “positive” wells to obtain a “fixed” number would also result in bias introduction. For this reason, the only sensible option would be to use the current average for each concentration positive wells (i.e., range of…) and apply the resulting pEC50 values (from three independent experiments performed on different days and with different cell bathes and transfections) to perform the ANOVA analyses. To be clear, the ANOVAs were not determined by a range of varying numbers of a datapoint but by the results obtained from three independent experiments.